# IRIS: Implicit Reward-Guided Internal Sifting
# for Mitigating Multimodal Hallucination

Yuanshuai Li[1]   Yuping Yan[1]   Jirui Han[1]   Fei Ming[1]   Lingjuan Lyu[2]   Yaochu Jin[1][†]

## Abstract

Hallucination remains a fundamental challenge for Multimodal Large Language Models (MLLMs). While Direct Preference Optimization (DPO) is a key alignment framework, existing approaches often rely heavily on costly external evaluators for scoring or rewriting, incurring off-policy learnability gaps and discretization loss. Due to the lack of access to internal states, such feedback overlooks the fine-grained conflicts between different modalities that lead to hallucinations during generation. To address this issue, we propose **IRIS** (**I**mplicit **R**eward-Guided **I**nternal **S**ifting), which leverages continuous implicit rewards in the native log-probability space to preserve fine-grained preference information and capture internal modal competition. After an SFT warm-up, IRIS performs on-policy preference alignment by sifting self-generated responses sampled from the current policy. These responses are then ranked with multimodal implicit rewards to form preference pairs that drive optimization toward resolving modal conflicts. Extensive experiments demonstrate that IRIS achieves highly competitive performance on key hallucination benchmarks using only 5.7k samples, without requiring any external feedback during preference alignment. These results confirm that IRIS provides an efficient and principled paradigm for mitigating MLLM hallucinations. Code is available here.

## 1. Introduction

Multimodal Large Language Models (MLLMs) (Achiam et al., 2023; Liu et al., 2023) integrate pretrained visual encoders with Large Language Models (LLMs) to achieve

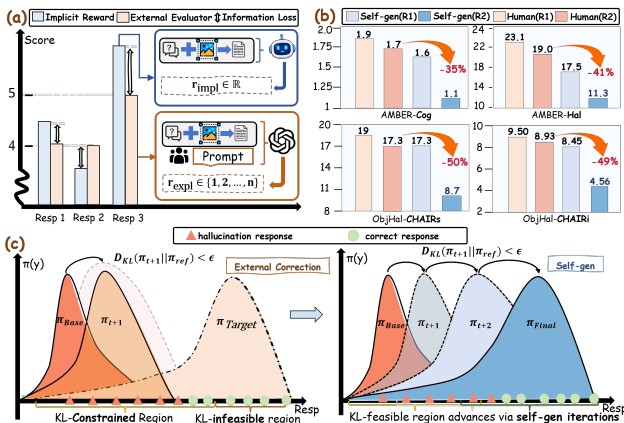

*Figure 1.* **Overview of IRIS. (a)** Comparison between external discrete rewards and IRIS implicit rewards. **(b)** Empirical hallucination reduction achieved by self-generated preference optimization compared to human-annotated data under the same experimental setting. **(c)** Conceptual illustration of policy evolution under KL constraints: external correction fails to cross the KL-infeasible region, while IRIS progressively advances the KL-feasible region via self-generated iterations.

strong performance on vision-language tasks. However, these models often suffer from hallucinations, where the generated text contradicts the provided visual evidence (Liu et al., 2024). Fundamentally, hallucinations arise from an imbalance between modalities during the generation process. MLLMs exhibit a strong dependence on statistical language priors acquired from large-scale textual pretraining (Leng et al., 2025). These priors can dominate the influence of visual signals during generation, leading the model to prioritize linguistically plausible responses that are insufficiently grounded in visual evidence (Leng et al., 2024a). Consequently, the model fails to faithfully ground its generation in visual inputs, especially when visual information contradicts common linguistic patterns.

To better align MLLMs with visual evidence, Direct Preference Optimization (DPO) (Rafailov et al., 2023) offers a rigorous objective for preference alignment by directly optimizing the policy on preference pairs without explicit reward modeling. This objective enables stable optimization under KL-divergence constraints (Kullback & Leibler, 1951) by establishing a mapping between the reward func-

[1]Department of Artificial Intelligence, Westlake University, Hangzhou, China [2]Sony AI, Zurich, Switzerland. Correspondence to: Yaochu Jin <jinyaochu@westlake.edu.cn>.

*Proceedings of the 43rd International Conference on Machine Learning*, Seoul, South Korea. PMLR 306, 2026. Copyright 2026 by the author(s).

tion and the optimal policy. Despite its widespread adoption, the efficacy of DPO in mitigating multimodal hallucinations is highly dependent on whether preference signals can capture the degree of visual grounding in the model's own generative process.

This leads to a fundamental question: which preference signals can reliably quantify the degree of visual grounding during the generation process? Existing approaches predominantly utilize external evaluators, such as GPT-4V, to provide discrete scores or corrective feedback (Yu et al., 2024b; Yang et al., 2025; Liu et al., 2025; Li et al., 2025a). However, these signals primarily assess output semantics and fail to characterize the specific internal mechanisms responsible for hallucinations along the model's own generative trajectories, leading to two key limitations. **First,** external evaluators induce information loss through discretization. They compress the model's continuous probability distribution into a restricted set of labels. As illustrated in Fig. 1a, discrete external rewards assign identical scores to semantically distinct responses, collapsing fine-grained preference differences that are essential for accurate visual grounding (Wang et al., 2025b). **Second,** external preference signals introduce a structural distributional discrepancy that hinders optimization. The reverse KL-divergence constraint in the DPO objective confines policy refinement to the support of the reference distribution. When preferred responses originate from disjoint off-policy distributions, they receive vanishing probability under the reference policy, so the corresponding log-ratio terms contribute almost no effective gradient and the update weights approach zero, rendering the feedback unlearnable (Guo et al., 2024), as illustrated by the target lying beyond the KL-constrained region in Fig. 1c (left).

In contrast to external evaluators, DPO leverages an *implicit reward* that is defined from the model's own policy (Rafailov et al., 2023). It is the log-likelihood ratio between the current and reference policies, and it provides a continuous signal in log-probability space. Compared with discrete feedback, it retains fine-grained preference differences that discretization would discard. Furthermore, after an SFT warm-up for visual consistency (Wang et al., 2025a), applying this signal to self-generated candidates helps reduce distribution shift by keeping preference construction within the model's own generation distribution. This facilitates stable learning under KL constraints and helps identify when language priors override visual evidence, supporting iterative on-policy refinement that improves visual grounding and reduces hallucinations.

Inspired by these insights, we introduce **IRIS** (**I**mplicit **R**eward-Guided **I**nternal **S**ifting), a multimodal preference alignment framework that leverages intrinsic implicit rewards as the primary alignment signal. By reducing the dependency on external evaluators during preference alignment, IRIS enables the model to autonomously refine its policy using its native implicit rewards (Fig. 1a). The framework operates in two stages. First, a preliminary SFT phase is conducted for value calibration, anchoring the model's latent distribution to visual consistency. Building on this foundation, the model first generates candidate responses under its current policy; these candidates serve as preference answers, while implicit rewards with Rectified Visual Guidance (RVG) serve as the preference signal for sifting them into on-policy preference pairs. These pairs are subsequently optimized via multimodal DPO in an iterative refinement cycle, ensuring that the alignment remains grounded in the model's intrinsic generative distribution (Fig. 1c).

Our primary contributions are summarized as follows:

- We identify the limitations of discrete external feedback in multimodal DPO, and show that implicit rewards provide a continuous signal better suited for mitigating hallucinations.
- We propose **IRIS**, an efficient and principled paradigm that leverages Rectified Visual Guidance (RVG) scoring to sift on-policy preference pairs, ensuring the alignment is grounded in the model's native distribution.
- With only 5.7k samples and no external feedback during alignment, IRIS achieves strong and competitive performance across multiple benchmarks, matching or outperforming baselines trained with substantially larger datasets and external evaluators on key hallucination metrics.

## 2. Related Work

### 2.1. Hallucination Mitigation in MLLMs

Research on mitigating hallucinations in MLLMs has evolved from inference-time decoding strategies (Leng et al., 2024b; Chen et al., 2024b) toward training-time preference alignment. Recently, DPO-based approaches have become the prevailing paradigm for enhancing visual grounding, primarily due to their superior stability and computational efficiency compared to traditional Reinforcement Learning from Human Feedback (RLHF) frameworks (Yu et al., 2024a) that rely on complex and often unstable optimization procedures like PPO (Schulman et al., 2017).

The efficacy of DPO alignment depends on the quality of preference pairs, which provide the essential supervision to distinguish grounded responses from hallucinations. Early research focused on ranking-based automated exploration. These methods establish heuristic rules to construct preference pairs, employing heuristic metrics such as cross-modal similarity (Ouali et al., 2024), model scaling priors (Zhang et al., 2024), or visual input perturbations (Pi et al., 2024)

to estimate response quality.

To achieve higher alignment precision, the focus has shifted toward expert-led feedback. This progression has moved from fine-grained human annotations (Yu et al., 2024a) to leveraging proprietary models like GPT-4V (Yang et al., 2025; Liu et al., 2025), and more recently, to utilizing powerful open-source models (Yu et al., 2024b; Liu et al., 2025) as external evaluators for hallucination detection and rewriting.

Despite these differences, these approaches rely on an external supervision paradigm and typically utilize off-policy data. Consequently, they are limited by the capabilities of the evaluators and fail to address the internal causes of hallucinations during the model's own generation process.

### 2.2. Self-Alignment via DPO Implicit Rewards

Although DPO and its variants are widely adopted for their simplicity, their offline nature can induce distribution shift, limiting policy improvement and potentially leading to overfitting (Guo et al., 2024). Prior work suggests that incorporating on-policy sampling to provide dynamic feedback significantly enhances alignment stability and performance (Tajwar et al., 2024). Consequently, mining and filtering self-generated samples for self-alignment has emerged as a key strategy to overcome the inherent limitations of offline DPO.

Theoretical evidence supports this internal evaluation approach. Rafailov et al. (2024) showed that DPO-trained models implicitly define a dense reward function at the token level. Recent research further confirms that models have the potential to evaluate themselves. For instance, it is found in Li et al. (2025c) that models possess internal rewards for self-evaluation, while authors in Wang et al. (2025a) showed that SFT helps calibrate these reward signals. Based on these findings, textual alignment methods such as DICE (Chen et al., 2024a) and SeRA (Ko et al., 2025) have been proposed to leverage implicit reward signals for general quality improvement through sample bootstrapping and filtering.

However, the potential of implicit rewards for multimodal hallucination mitigation beyond text-only self-alignment remains unexplored. We argue that this signal is naturally suited for this task because it directly reflects the internal competition between visual evidence and language priors. This allows us to detect hallucinations within the model's own probability space, which is not possible with external evaluators that only observe final outputs.

## 3. Preliminaries

We formalize the problem of multimodal hallucination mitigation within the framework of preference optimization. Let $v$ denote the visual input (image), $x$ the textual instruction, and $y = (y_1, \ldots, y_T)$ the generated response sequence.

**Supervised Fine-Tuning (SFT)** SFT is a widely adopted technique to adapt pre-trained MLLMs to downstream tasks. Given a dataset $\mathcal{D}_{\text{SFT}} = \{(v, x, y)\}$ comprising visual inputs $v$, textual prompts $x$, and corresponding ground-truth responses $y$, the training objective is to maximize the negative likelihood of the target response in an auto-regressive manner:

$$\mathcal{L}_{\text{SFT}}(\pi_\theta) = -\mathbb{E}_{(v,x,y)} \left[ \sum_{t=1}^{|y|} \log \pi_\theta(y_t \mid v, x, y_{<t}) \right], \quad (1)$$

where $y_t$ denotes the $t$-th token in the target sequence, $y_{<t}$ represents the sequence of preceding tokens, and $\pi_\theta(y_t \mid v, x, y_{<t})$ is the conditional probability predicted by the model.

**Direct Preference Optimization (DPO).** DPO optimizes a policy model using paired preference data $\mathcal{D}_{\text{pref}} = \{(v, x, y_w, y_l)\}$, where $y_w$ is preferred over $y_l$ given the same input $(v, x)$. Under the Bradley-Terry (BT) model assumption, the preference probability is determined by a latent reward function $r^*$:

$$P(y_w \succ y_l \mid v, x) = \sigma(r^*(v, x, y_w) - r^*(v, x, y_l)), \quad (2)$$

where $\sigma$ denotes the sigmoid function. DPO derives a closed-form mapping between the optimal reward function $r^*$ and the optimal policy $\pi^*$. Specifically, the implicit reward is expressed as:

$$r^*(v, x, y) = \beta \log \frac{\pi^*(y \mid v, x)}{\pi_{\text{ref}}(y \mid v, x)} + Z(v, x), \quad (3)$$

where $\beta$ controls the deviation from the reference policy and $Z(v, x)$ is a partition function. Based on this formulation, the policy $\pi_\theta$ is optimized by minimizing the following objective:

$$\mathcal{L}_{\text{DPO}}(\pi_\theta) = -\mathbb{E}_{(v,x,y_w,y_l)} \left[ \log \sigma \left( \beta \log \frac{\pi_\theta(y_w \mid v, x)}{\pi_{\text{ref}}(y_w \mid v, x)} \right. \right.$$
$$\left. \left. - \beta \log \frac{\pi_\theta(y_l \mid v, x)}{\pi_{\text{ref}}(y_l \mid v, x)} \right) \right].$$
$$(4)$$

This formulation unifies reward modeling and policy optimization into a single objective, eliminating the need for an explicit reward model typically required in reinforcement learning.

## 4. Methodology

### 4.1. Theoretical Motivation

**Learnability of Noisy Self-Generated Preferences.** A potential concern is that self-generated candidates $(y_w, y_l)$

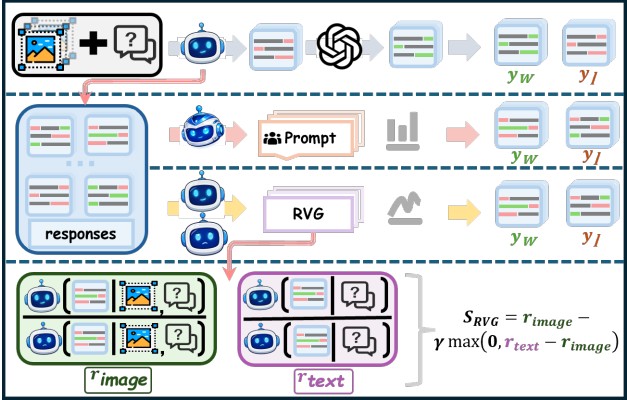

*Figure 2.* **Comparison of Preference Construction Pipelines. Top:** Feedback from proprietary models (e.g., GPT-4). **Middle:** Prompt-based scoring using large open-source models. **Bottom:** IRIS (Ours), which leverages intrinsic implicit rewards and Rectified Visual Guidance (RVG) to sift on-policy preference pairs without external evaluators.

may be imperfect and even share hallucinated content. Preference optimization, however, depends on *relative* comparisons rather than absolute correctness. Under the delta learning view (Geng et al., 2025), learning can still progress as long as the induced preference direction is correct more often than not.

For a generic pairwise objective, the gradient admits a difference form:

$$\nabla_\theta \mathcal{L}_{\text{pref}} = - w(\cdot)\Big(\nabla_\theta \log \pi_\theta(y_w \mid v, x) - \nabla_\theta \log \pi_\theta(y_l \mid v, x)\Big), \quad (5)$$

where $w(\cdot) > 0$ is a scalar weight determined by the loss. Thus, the update increases the log-likelihood gap $\log \pi_\theta(y_w \mid v, x) - \log \pi_\theta(y_l \mid v, x)$ in expectation over sampled preference pairs, pushing the policy toward responses that are preferred under the same context. A formal derivation is provided in Appendix H.

**Implicit Reward Calibration via SFT.** Self-alignment with implicit rewards requires the model to rank grounded responses above hallucinated ones. We posit that this capability is established during the SFT warm-up, which calibrates the model's scoring toward task-relevant visual grounding. Recent analyses relate maximum-likelihood training to implicit reward learning under KL-regularized distribution matching (Wang et al., 2025a).

In particular, under an implicit-reward formulation, the soft-optimal policy admits

$$\log \pi^*(a \mid s) = \log \pi_{\text{ref}}(a \mid s) + \frac{1}{\beta} Q^*(s, a) - V^*(s), \quad (6)$$

and hence $\log \frac{\pi^*(a|s)}{\pi_{\text{ref}}(a|s)} = \frac{1}{\beta} Q^*(s, a) - V^*(s)$. Motivated by

this connection, we use the DPO implicit reward $r(y \mid s) = \beta \log \frac{\pi(y|s)}{\pi_{\text{ref}}(y|s)}$ as an intrinsic scoring signal.

We treat the generation context as the state: $s = (v, x)$ for image-conditioned generation and $s = (v_\emptyset, x)$ for text-only generation. Accordingly, we define $r_{\text{image}} = r(y \mid v, x)$ and $r_{\text{text}} = r(y \mid v_\emptyset, x)$. Comparing $r_{\text{image}}$ and $r_{\text{text}}$ for the same candidate $y$ separates visual evidence from language-only priors; in particular, $r_{\text{text}} > r_{\text{image}}$ indicates that $y$ is more strongly supported under the text-only context, which motivates the rectification in preference construction.

### 4.2. Warm-up and On-policy Self-Generation

We start from a base model $\pi_{\theta_{\text{base}}}$ and perform an SFT warm-up on $\mathcal{D}_{\text{SFT}}$ to obtain a visually grounded instruction-following policy $\pi_{\theta_0}$, which calibrates the initial implicit reward landscape.

IRIS then proceeds in iterative preference rounds indexed by $r = 1, 2, \ldots$ as illustrated in Figure 2. In round $r$, for each $(v, x) \in \mathcal{D}_{\text{SFT}}$, we sample $K$ candidate responses from the previous-round policy $\pi_{\theta_{r-1}}$:

$$\{y^{(k)}\}_{k=1}^K \sim \pi_{\theta_{r-1}}(\cdot \mid v, x). \quad (7)$$

These self-generated responses are subsequently scored and sifted to form on-policy preference pairs.

### 4.3. Implicit Reward Scoring and Preference Pair Construction

To construct on-policy preference pairs, we score each sampled response $y$ using length-normalized log-likelihood ratios between the sampling policy $\pi_{\theta_{r-1}}$ and the *scoring reference* $\pi_{\text{ref}}^{(r-1)}$, taken as the preceding policy $\pi_{\theta_{r-2}}$. These ratios are utilized exclusively for scoring and sifting samples into preference pairs.

The image-conditioned implicit reward, which captures grounded alignment under the visual context $v$, is defined as:

$$r_{\text{image}}^{(r)}(v, x, y) = \log \frac{\pi_{\theta_{r-1}}(y \mid v, x)}{\pi_{\text{ref}}^{(r-1)}(y \mid v, x)}. \quad (8)$$

The text-only implicit reward, which isolates language priors by omitting the visual context via $v_\emptyset$, is defined as:

$$r_{\text{text}}^{(r)}(x, y) = \log \frac{\pi_{\theta_{r-1}}(y \mid v_\emptyset, x)}{\pi_{\text{ref}}^{(r-1)}(y \mid v_\emptyset, x)}. \quad (9)$$

To eliminate the influence of response length on the implicit rewards, the log-likelihood is normalized by the token count $|y|$ as $\log \pi(y \mid \cdot) = \frac{1}{|y|} \sum_{t=1}^{|y|} \log \pi(y_t \mid \cdot, y_{<t})$. This normalization prevents the scoring process from being biased by sequence length, ensuring that the selection is solely determined by the grounding quality of the candidates.

We then define a comprehensive grounding-aware score using **Rectified Visual Guidance (RVG)**:

$$S^{(r)}(v, x, y) = r^{(r)}_{\text{image}}(v, x, y)$$
$$- \gamma \max\left(0, r^{(r)}_{\text{text}}(x, y) - r^{(r)}_{\text{image}}(v, x, y)\right), \tag{10}$$

where $\gamma \geq 0$ is the rectification strength. The penalty term is activated only when $r^{(r)}_{\text{text}} > r^{(r)}_{\text{image}}$, which corresponds to cases where the model assigns a higher relative likelihood to a response in the absence of visual evidence. This behavior is attributed to an over-reliance on language priors and is identified as a primary source of multimodal hallucinations (Xie et al., 2024). By down-weighting such candidates, RVG enforces a grounding constraint that prioritizes responses derived from actual visual input.

Given the $K$ candidates $\{y^{(k)}\}_{k=1}^K$ sampled for each input $(v, x)$, we identify the responses with the maximum and minimum $S^{(r)}$ scores to construct an on-policy preference pair with the highest contrast. This sifting process ensures that the optimization is guided by the most distinct supervisory signals available within the sampled set:

$$y_w = \arg \max_{k \in \{1, \ldots, K\}} S^{(r)}(v, x, y^{(k)}),$$
$$y_l = \arg \min_{k \in \{1, \ldots, K\}} S^{(r)}(v, x, y^{(k)}). \tag{11}$$

In round $r$, we designate $y_w$ as the preferred response and $y_l$ as the rejected counterpart for preference optimization.

To ensure high-quality preference data, we apply a lightweight filtering stage with degeneration screening and length-aware filtering to reduce noisy supervision and length bias. For samples that would otherwise be discarded, we recover them by anchoring the chosen response to the SFT reference from $\mathcal{D}_{\text{SFT}}$. This improves training stability across rounds while keeping the pipeline self-contained without external evaluators. We provide the full filtering rules and anchoring details in Appendix C.

### 4.4. Grounded Preference Learning Objectives

Based on the preference pairs identified through the scoring and sifting process, we update the policy to improve multimodal grounding and alignment. In round $r$, we optimize parameters $\theta$ initialized from $\theta_{r-1}$ and denote the resulting policy as $\pi_{\theta_r}$; for notational simplicity, we write $\pi_\theta$ for the policy being optimized. We minimize a composite objective that combines Conditional Textual Preference, Conditional Visual Preference, and Anchored Regularization. In each round $r$, the *DPO reference* $\pi_{\text{ref}}$ is a frozen copy of the preceding policy $\pi_{\theta_{r-1}}$.

**Conditional Textual Preference** The component $\mathcal{L}_{\text{ctp}}$ adopts the standard DPO objective on $\mathcal{D}^{(r)}$, increasing the preference margin of $y_w$ over $y_l$ under the multimodal context $(v, x)$ relative to the frozen reference policy $\pi_{\text{ref}}$:

$$\mathcal{L}_{\text{ctp}} = -\mathbb{E}_{(v, x, y_w, y_l) \sim \mathcal{D}^{(r)}} \Big[ \log \sigma \big( \beta \log \frac{\pi_\theta(y_w \mid v, x)}{\pi_{\text{ref}}(y_w \mid v, x)}$$
$$- \beta \log \frac{\pi_\theta(y_l \mid v, x)}{\pi_{\text{ref}}(y_l \mid v, x)} \big) \Big]. \tag{12}$$

This term serves as the core preference-learning signal on the sifted on-policy pairs.

**Conditional Visual Preference** The component $\mathcal{L}_{\text{cvp}}$ encourages visual dependence by preferring the same response $y_w$ under the original image $v$ over a perturbed counterpart $\tilde{v}$ where the evidence supporting $y_w$ is suppressed:

$$\mathcal{L}_{\text{cvp}} = -\mathbb{E}_{(v, x, y_w) \sim \mathcal{D}^{(r)}} \Big[ \log \sigma \big( \beta \log \frac{\pi_\theta(y_w \mid v, x)}{\pi_{\text{ref}}(y_w \mid v, x)}$$
$$- \beta \log \frac{\pi_\theta(y_w \mid \tilde{v}, x)}{\pi_{\text{ref}}(y_w \mid \tilde{v}, x)} \big) \Big]. \tag{13}$$

Here, $\tilde{v}$ is generated by applying a perturbation operator $T(\cdot)$ to $v$. This term discourages high relative reward for $y_w$ under visually uninformative inputs, thereby promoting grounded preference learning.

**Anchored Regularization** The component $\mathcal{L}_{\text{anchor}}$ stabilizes training by preventing the likelihood of preferred responses from drifting downward. Since DPO-style objectives enforce only a *relative* margin between $(y_w, y_l)$, the preference gap can increase even if the likelihood of $y_w$ decreases. We therefore introduce an anchored term (Wang et al., 2024; Yang et al., 2025; Liu et al., 2025) that keeps the reference-relative reward of $y_w$ above a soft margin $\delta$:

$$\mathcal{L}_{\text{anchor}} = -\mathbb{E}_{(v, x, y_w) \sim \mathcal{D}^{(r)}} \Big[ \log \sigma \big( \beta \log \frac{\pi_\theta(y_w \mid v, x)}{\pi_{\text{ref}}(y_w \mid v, x)}$$
$$- \delta \big) \Big]. \tag{14}$$

Here, $\delta$ specifies a soft margin on the reference-relative reward of $y_w$.

**Total Objective** By combining these components, the total objective is defined as the weighted sum of the grounded learning signals:

$$\mathcal{L}_{\text{total}} = \mathcal{L}_{\text{ctp}} + \lambda \mathcal{L}_{\text{cvp}} + \mathcal{L}_{\text{anchor}}, \tag{15}$$

where $\lambda$ controls the strength of $\mathcal{L}_{\text{cvp}}$.

**Iterative Alignment with Separate References.** IRIS starts from a base model $\pi_{\theta_{\text{base}}}$ and performs an SFT warm-up to obtain $\pi_{\theta_0}$. For each preference round, we use two references for different purposes. When constructing the

on-policy preference set, we score self-generated candidates using implicit reward ratios computed between two consecutive policies: in round $r$, scoring uses $\pi_{\theta_{r-1}}$ with $\pi_{\theta_{r-2}}$ as the reference for the log-ratio. For $r = 1$, we use the base model $\pi_{\theta_{\text{base}}}$ as the scoring reference. During preference optimization, we initialize the trainable policy from the previous round and use a frozen copy of it as the DPO reference within the round, namely $\pi_\theta \leftarrow \pi_{\theta_{r-1}}$ and $\pi_{\text{ref}} \leftarrow \pi_{\theta_{r-1}}$. We then minimize Eq. 15 on $\mathcal{D}^{(r)}$ to obtain the updated policy $\pi_{\theta_r}$, and repeat for a small number of rounds.

## 5. Experiments

### 5.1. Experimental Setup

**Implementation Details.** IRIS is implemented on LLaVA-1.5 7B and 13B. The models use CLIP ViT-L/14 and Vicuna-v1.5 as backbones. We set $\gamma = 0.7$ for RVG and use $K = 5$ for on-policy sampling. The generation temperature is 0.7. SFT warm-up uses 5,700 samples from the RLHF-V dataset. For training, we set $\beta = 0.1$ and $\lambda = 1.0$ for $\mathcal{L}_{\text{cvp}}$. The 7B and 13B models use learning rates of $5 \times 10^{-7}$ and $1 \times 10^{-6}$, respectively. Each round is trained for 2 epochs. All experiments are run on 8 NVIDIA H20 GPUs. The preference alignment stage operates without external human or AI feedback. Details on constructing the rejected images $\tilde{v}$ are provided in Appendix C.

**Evaluation Benchmarks.** We evaluate IRIS on several representative benchmarks to assess both hallucination mitigation and general capabilities. **AMBER** (Wang et al., 2023) is a multi-dimensional generative benchmark with 1,004 images; we report object hallucination (**CHAIR↓**), coverage (**Cover↑**), response-level hallucination (**HalRate↓**), and cognitive hallucination (**Cog↓**). **MMHal-Bench** (Sun et al., 2024) includes 96 images across 12 categories for question answering; we follow the official rubric to report the overall quality **Score↑** and **HalRate↓** using GPT-4 evaluation. **Object-HalBench** (Rohrbach et al., 2018) consists of 300 instances for image description; we report CHAIR metrics at both the sentence (**CHAIRs↓**) and instance levels (**CHAIRi↓**). Finally, **LLaVA-Bench (in-the-Wild)** (Liu et al., 2023) is used to assess general conversational ability via GPT-4-relative scores.

**Baselines.** We compare IRIS against a comprehensive set of recent state-of-the-art approaches for hallucination mitigation, including **LLaVA-RLHF** (Sun et al., 2024), **HALVA** (Sarkar et al., 2024), **mDPO** (Wang et al., 2024), **HA-DPO** (Zhao et al., 2023), **V-DPO** (Xie et al., 2024), **POVID** (Zhou et al., 2024), **RLAIF-V** (Yu et al., 2024b), **SymMPO** (Liu et al., 2025), **RLHF-V (HD)** (Yu et al., 2024a), **LPOI** (Zadeh et al., 2025), and **OPA-DPO** (Yang et al., 2025).

### 5.2. Main Results

Table 1 summarizes the main results on three representative hallucination benchmarks. We report IRIS-R2 as our final model. Overall, IRIS-R2 improves grounding-oriented performance on both 7B and 13B backbones. On AMBER, it reduces hallucination-related metrics such as CHAIR and Hal-Rate compared to the vanilla LLaVA-1.5 models, while keeping coverage at a similar level. On MMHal-Bench, IRIS-R2 also improves over the base models, but the gains are smaller than those on object-level hallucination metrics; this may be partly because MMHal-Bench emphasizes compositional visual reasoning (e.g., counting and relations), while our method focuses on improving visual grounding to reduce hallucinated objects and attributes. Notably, on Object HalBench, IRIS-R2 achieves **8.66** CHAIRs, showing strong improvements in fine-grained object grounding. Furthermore, we observe that IRIS-R2 consistently outperforms IRIS-R1 across model scales, validating the effectiveness of iterative self-alignment.

We further compare IRIS with two recent strong baselines, RLAIF-V (Yu et al., 2024b) and OPA-DPO (Yang et al., 2025). Compared to RLAIF-V, IRIS is competitive on AMBER and achieves strong object-level grounding on Object HalBench. Compared to OPA-DPO, which relies on GPT-4V feedback, IRIS remains competitive on AM-BER while achieving clear gains on Object HalBench and attaining higher coverage. Crucially, regarding efficiency, while RLAIF-V also employs open-source models, it relies on heavy prompt-based labeler scoring. In contrast, IRIS leverages implicit rewards, resulting in substantially lower curation cost under our measured setting (Appendix B).

### 5.3. Ablation Studies

**Effect of Objective Components.** Table 2 isolates the effect of each objective component. The results show that conditional visual preference, denoted by $\mathcal{L}_{\text{cvp}}$, substantially improves CHAIR-based hallucination metrics. Conditional visual preference is the main signal for grounding, while anchored regularization, $\mathcal{L}_{\text{anchor}}$, helps stabilize training and prevent capability degradation. Removing anchored regularization leads to a drop in general capability below the vanilla base model, as further evidenced in Table 9

**Impact of Training Paradigms.** Table 3 studies two factors in our training pipeline: **SFT warm-up** and on-policy **self-generation**. We first find that the SFT warm-up gives a clearly better starting point: with the same training round, models with SFT warm-up generally show lower hallucination metrics than those trained without it. This suggests that SFT helps the policy learn a more grounded response pattern before preference optimization.

More importantly, on-policy self-generation brings the

*Table 1.* Comparative assessment of IRIS against state-of-the-art baselines on multimodal hallucination benchmarks. Boldface indicates the best result. Values in parentheses denote the absolute change with respect to the corresponding vanilla LLaVA-1.5 backbone.

| | | | AMBER | | | | MMHAL | | OBJECT HAL | |
|---|---|---|---|---|---|---|---|---|---|---|
| ALGORITHM | DATA SIZE | FEEDBACK | CHAIR↓ | COVER↑ | HALRATE↓ | COG↓ | SCORE↑ | HALRATE↓ | CHAIRs↓ | CHAIRi↓ |
| GPT-4V (YANG ET AL., 2023) | × | × | 4.6 | 67.1 | 30.7 | 2.6 | 3.49 | 0.28 | 13.6 | 7.3 |
| QWEN-VL-CHAT (BAI ET AL., 2023) | × | × | 6.6 | 53.2 | 31.0 | 2.9 | 2.89 | 0.43 | 36.0 | 21.3 |
| SILKIE (LI ET AL., 2023) | × | × | 5.4 | 55.8 | 29.0 | 2.0 | 3.01 | 0.41 | 25.3 | 13.9 |
| INSTRUCTBLIP (DAI ET AL., 2023) | × | × | 8.8 | 52.2 | 38.2 | 4.4 | 2.14 | 0.58 | 25.9 | 14.3 |
| MINIGEMINI (LI ET AL., 2025B) | × | × | - | - | - | - | 3.08 | 0.38 | 14.5 | 8.0 |
| **LLAVA-1.5-7B (LIU ET AL., 2023)** | | | 8.8 | 50.1 | 40.4 | 4.7 | 2.18 | 0.59 | 54.70 | 26.5 |
| +LLAVA-RLHF (SUN ET AL., 2024) | 122K | RLHF | 9.7 | **53.2** | 46.6 | 5.3 | 1.88 | 0.71 | 58.00 | 15.61 |
| +HALVA (SARKAR ET AL., 2024) | 21.5K | GPT-4V | 6.6 | 53.0 | 32.2 | 3.4 | 2.25 | 0.54 | 41.40 | 11.70 |
| +MDPO (WANG ET AL., 2024) | 10K | GPT-4V | 4.4 | 52.4 | 24.5 | 2.4 | 2.39 | 0.54 | 35.70 | 9.80 |
| +HA-DPO (ZHAO ET AL., 2023) | 6K | GPT-4 | 7.8 | 52.1 | 35.6 | 4.2 | 1.89 | 0.65 | 54.00 | 14.45 |
| +V-DPO (XIE ET AL., 2024) | 10K | GPT-3.5 | 6.6 | 49.1 | 30.8 | 3.1 | - | - | - | - |
| +POVID (ZHOU ET AL., 2024) | 17K | GPT-4V | 7.4 | 51.3 | 34.3 | 3.9 | 2.08 | 0.60 | 50.67 | 15.28 |
| +RLAIF-V (YU ET AL., 2024B) | 16K | LLAVA-NEXT | 3.0 | 50.4 | 16.2 | 1.0 | **3.00** | **0.38** | 16.00 | **3.70** |
| +SYMMPO (LIU ET AL., 2025) | 21K | DEEPSEEK-V3 | 5.2 | 49.5 | 25.4 | 3.0 | 2.63 | 0.51 | 20.4 | 10.3 |
| +OPA-DPO (YANG ET AL., 2025) | 4.8K | GPT-4V | **2.2** | 47.9 | 11.6 | **0.9** | 2.83 | 0.45 | 13.00 | 4.25 |
| +LPOI (ZADEH ET AL., 2025) | 10K | GPT-4V | 4.3 | 51.9 | 26.4 | 2.0 | 2.40 | 0.59 | 24.3 | 14.6 |
| **+IRIS-R1 (OURS)** | 5.7K | IMPLICIT REWARD | 3.8(-5.0) | 51.9(+1.8) | 17.5(-22.9) | 1.6(-3.1) | 2.34(+0.16) | 0.50(-0.09) | 17.3(-37.4) | 8.45(-18.05) |
| **+IRIS-R2 (OURS)** | 5.7K | IMPLICIT REWARD | 2.4(-6.4) | 51.1(+1.0) | 11.3(-29.1) | 1.1(-3.6) | 2.42(+0.24) | 0.50(-0.09) | **8.66(-46.04)** | 4.56(-21.94) |
| **LLAVA-1.5-13B (LIU ET AL., 2023)** | | | 8.8 | 50.3 | 37.2 | 4.3 | 2.31 | 0.55 | 49.3 | 23.9 |
| +LLAVA-RLHF (SUN ET AL., 2024) | 122K | RLHF | 7.7 | 52.3 | 38.6 | 4.0 | 2.27 | 0.64 | 44.67 | 11.83 |
| +MDPO (WANG ET AL., 2024) | 10K | GPT-4V | 4.6 | 52.6 | 25.0 | 2.0 | 2.50 | 0.57 | 33.3 | 16.6 |
| +RLHF-V (HD) (YU ET AL., 2024A) | 1.4K | HUMAN | 6.3 | 46.1 | 25.1 | 2.1 | 2.81 | 0.49 | - | - |
| +HALVA (SARKAR ET AL., 2024) | 21.5K | GPT-4V | 6.4 | 52.6 | 30.4 | 3.2 | 2.58 | 0.45 | 45.40 | 12.80 |
| +SYMMPO (LIU ET AL., 2025) | 21K | DEEPSEEK-V3 | 4.8 | 52.8 | 25.1 | 2.1 | 2.85 | 0.48 | 18.3 | 10.0 |
| +LPOI (ZADEH ET AL., 2025) | 10K | GPT-4V | 3.9 | 52.9 | 22.3 | 1.8 | 2.54 | 0.57 | 24.3 | 11.7 |
| **+IRIS-R1 (OURS)** | 5.7K | IMPLICIT REWARD | 3.7(-5.1) | **53.7(+3.4)** | 20.2(-17.0) | 1.9(-2.4) | 2.82(+0.51) | 0.42(-0.14) | 18.6(-30.7) | 9.1(-14.8) |
| **+IRIS-R2 (OURS)** | 5.7K | IMPLICIT REWARD | 3.5(-5.3) | 52.2(+1.9) | 18(-19.2) | 1.7(-2.6) | **2.86(+0.55)** | **0.41(-0.14)** | 10(-39.3) | 5.49(-18.41) |

*Table 2.* Ablation study on objective components. Starting from $\mathcal{L}_{\text{ctp}}$, we add $\mathcal{L}_{\text{cvp}}$ and $\mathcal{L}_{\text{anchor}}$. The full objective achieves the best overall results, while $\mathcal{L}_{\text{cvp}}$ provides the primary gains.

| COMPONENTS | | | AMBER | | | OBJECT HAL | |
|---|---|---|---|---|---|---|---|
| $\mathcal{L}_{\text{CTP}}$ | $\mathcal{L}_{\text{CVP}}$ | $\mathcal{L}_{\text{ANCHOR}}$ | CHAIR↓ | HALRATE↓ | COG↓ | CHAIRs↓ | CHAIRi↓ |
| ✓ | × | × | 5.8 | 30.7 | 2.0 | 18.0 | 7.98 |
| ✓ | × | ✓ | 4.9 | 25.4 | 2.2 | 19.3 | 9.61 |
| ✓ | ✓ | × | 2.9 | **10.4** | **0.8** | 10.2 | 4.87 |
| ✓ | ✓ | ✓ | **2.4** | 11.3 | 1.1 | **8.66** | **4.56** |

*Table 3.* Factorized ablation on training paradigms. We evaluate the impact of **SFT warm-up** and on-policy **Self-gen** compared to off-policy human data (RLHF-V).

| Factors | | Round | AMBER | | | Object Hal | |
|---|---|---|---|---|---|---|---|
| SFT warm-up | Self-gen | | CHAIR↓ | HalRate↓ | Cog↓ | CHAIRs↓ | CHAIRi↓ |
| × | × | R0 | 5.9 | 29.8 | 3.3 | 43.3 | 21.3 |
| × | × | R1 | 4.9 | 25.3 | 2.8 | 31.0 | 15.3 |
| × | × | R2 | 3.8 | 20.2 | 2.1 | 27.3 | 13.0 |
| ✓ | × | R0 | 5.3 | 25.5 | 2.5 | 24.0 | 13.0 |
| ✓ | × | R1 | 4.6 | 23.1 | 1.9 | 19.0 | 9.5 |
| ✓ | × | R2 | 3.7 | 19.0 | 1.7 | 17.3 | 8.9 |
| × | ✓ | R0 | 5.9 | 29.8 | 3.3 | 43.3 | 21.3 |
| × | ✓ | R1 | 3.6 | 19.8 | 2.0 | 23.6 | 11.0 |
| × | ✓ | R2 | 2.5 | 14.4 | 1.5 | 18.3 | 9.22 |
| ✓ | ✓ | R0 | 5.3 | 25.5 | 2.5 | 24.0 | 13.0 |
| ✓ | ✓ | R1 | 3.8 | 17.5 | 1.6 | 17.3 | 8.45 |
| ✓ | ✓ | R2 | **2.4** | **11.3** | **1.1** | **8.6** | **4.56** |

largest improvements across rounds. When we replace static human pairs with self-generated pairs, the model reduces hallucinations faster and reaches a much better final result, especially on Object HalBench. In contrast, training with static human pairs yields more limited gains, which diminish as training proceeds. Overall, the results suggest that SFT warm-up mainly improves the initial model state, whereas on-policy self-generation enables sustained improvement across successive rounds.

**Effectiveness of Scoring Signals.** Table 4 compares alternative scoring signals for sifting preference pairs. In Round 1, the unimodal scores $r_{\text{text}}$ and $r_{\text{image}}$ yield comparable results and neither consistently dominates across metrics. In Round 2, $r_{\text{image}}$ improves upon $r_{\text{text}}$ on most hallucination measures, suggesting that visual conditioning becomes increasingly helpful as the policy is refined.

RVG performs best across all reported metrics. On Object HalBench, RVG reduces CHAIRs to 8.6 in Round 2, compared to 14.1 when sifting with $r_{\text{image}}$. This pattern is consistent with RVG suppressing candidates that remain

highly preferred under text-only priors but are weakly supported by the image, thereby producing more informative preference pairs for subsequent optimization. Appendix E further confirms the finer resolution of implicit rewards over external discrete scoring.

*Table 4.* Ablation on scoring signals for sifting. While using the image-only reward ($r_{\text{image}}$) is more effective than the text-only signal ($r_{\text{text}}$), our proposed RVG achieves the best performance.

| SCORING | ROUND | AMBER | | | OBJECT HAL | |
|---|---|---|---|---|---|---|
| | | CHAIR↓ | HALRATE↓ | COG↓ | CHAIRs↓ | CHAIRi↓ |
| $r_{\text{TEXT}}$ (TEXT-ONLY) | R1 | 3.6 | 17.9 | 1.8 | 17.9 | 9.22 |
| $r_{\text{TEXT}}$ (TEXT-ONLY) | R2 | 3.2 | 15.6 | 1.5 | 15.2 | 7.85 |
| $r_{\text{IMAGE}}$ (IMAGE-ONLY) | R1 | 3.8 | 17.8 | 1.8 | 18.3 | 9.30 |
| $r_{\text{IMAGE}}$ (IMAGE-ONLY) | R2 | 3.0 | 14.5 | 1.4 | 14.1 | 7.12 |
| RVG (OURS) | R1 | 3.8 | 17.5 | 1.6 | 17.3 | 8.45 |
| RVG (OURS) | R2 | **2.4** | **11.3** | **1.1** | **8.6** | **4.56** |

## 6. Hyperparameter Sensitivity

We examine the sensitivity of two key hyperparameters in IRIS: the rectification strength $\gamma$ in RVG, which controls the penalty for unsupported language priors, and the weight $\lambda$, which balances the conditional visual preference term. Figures 3 and 4 summarize the trends across a wide range of values. Overall, the performance is robust, with the optimal results achieved around $\gamma = 0.7$ and $\lambda = 1.0$. We adopt these as default settings in all subsequent experiments. Detailed numerical results and comprehensive sensitivity analyses are presented in Appendix G.

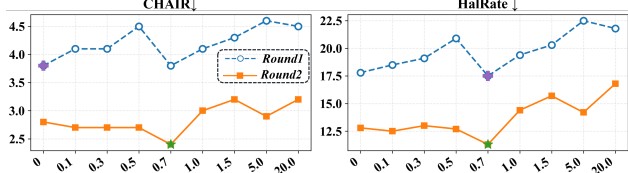

*Figure 3.* **Effect of Rectification Strength $\gamma$.** Sensitivity of hallucination metrics to $\gamma$ across two iterative rounds. The star indicates the optimal value at $\gamma = 0.7$.

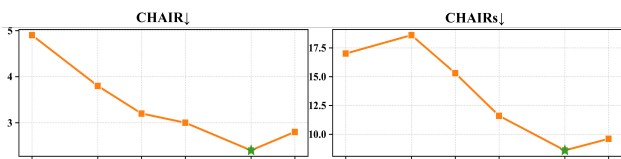

*Figure 4.* **Effect of Visual Preference Weight $\lambda$.** Sensitivity of hallucination metrics to the weight $\lambda$ in the final round. The star indicates the optimal value at $\lambda = 1.0$.

## 7. Further Analysis

**Data Efficiency.** Table 5 examines how IRIS changes with the amount of on-policy preference data. With only 1k pairs, IRIS already improves hallucination metrics on both AMBER and Object HalBench, compared to the model at the start of preference training. Increasing the budget to 3k pairs leads to much larger gains, and using 5.7k pairs gives the best or tied-best results on most metrics. Across all three budgets, Round 2 consistently outperforms Round 1, showing that iterative on-policy refinement remains effective without requiring tens of thousands of preference pairs.

*Table 5.* Data efficiency analysis. IRIS achieves strong performance with limited training data, demonstrating sample efficiency.

| DATA | ROUND | AMBER | | | OBJECT HAL | |
|---|---|---|---|---|---|---|
| | | CHAIR↓ | HALRATE↓ | COG↓ | CHAIRs↓ | CHAIRi↓ |
| 1K | R1 | 5.2 | 25.5 | 2.6 | 25.0 | 13.0 |
| 1K | R2 | 4.9 | 23.8 | 2.2 | 24.3 | 12.3 |
| 3K | R1 | 4.8 | 22.6 | 2.1 | 20.6 | 10.7 |
| 3K | R2 | 2.9 | 13.3 | **1.1** | 11.0 | 5.54 |
| 5.7K | R1 | 3.8 | 17.5 | 1.6 | 17.3 | 8.45 |
| 5.7K | R2 | **2.4** | **11.3** | **1.1** | **8.6** | **4.56** |

**Robustness to Sampling Repeat Times $K$.** Table 6 studies how sensitive IRIS is to the sampling repeat factor $K$. Performance stays similar across $K \in \{3, 5, 10\}$, and each setting improves from Round 1 to Round 2. The final results differ only a little, which suggests that IRIS does not need a large sampling budget to work well. Using more candidates can help, but the gains become smaller as $K$ increases. We set $K = 5$ as a simple default that gives strong results at a reasonable cost, and the method remains robust under other choices of $K$.

*Table 6.* Ablation on sampling repeat times $K$ for on-policy data generation.

| REPEAT $K$ | ROUND | AMBER | | | OBJECT HAL | |
|---|---|---|---|---|---|---|
| | | CHAIR↓ | HALRATE↓ | COG↓ | CHAIRs↓ | CHAIRi↓ |
| $K = 3$ | R1 | 3.6 | 16.7 | 1.7 | 17.6 | 9.28 |
| $K = 3$ | R2 | 2.8 | 12.1 | 1.2 | 7.66 | 4.10 |
| $K = 5$ | R1 | 3.8 | 17.5 | 1.6 | 17.3 | 8.45 |
| $K = 5$ | R2 | **2.4** | **11.3** | **1.1** | 8.6 | 4.56 |
| $K = 10$ | R1 | 3.6 | 16.3 | 1.5 | 16.3 | 7.75 |
| $K = 10$ | R2 | 2.7 | 12.2 | 1.5 | **6.0** | **3.55** |

## 8. Conclusion

We presented IRIS, an iterative on-policy self-alignment framework for mitigating hallucinations in MLLMs. The core of our approach is demonstrating that intrinsic implicit rewards can be effectively harnessed to identify high-quality preference signals from a model's own generative distribution, thereby reducing the dependency on costly external evaluators or proprietary models during preference alignment. By incorporating RVG during the sifting process, IRIS helps separate visual evidence from language priors,

enabling the model to refine its grounding through iterative refinement cycles.

Experimental results confirm that this paradigm consistently improves object-level grounding across multiple benchmarks. Our analysis further shows that `IRIS` is both sample-efficient and robust to hyperparameter choices, narrowing the performance gap to methods that rely on high-cost external feedback. Overall, by providing a practical and principled approach for internal preference mining, we believe `IRIS` offers a new and efficient perspective for mitigating hallucinations in future multimodal models.

## Acknowledgements

This work was supported by the International Collaboration Fund for Creative Research of the National Natural Science Foundation of China (NSFC ICFCRT) under Grant No. W2441019.

## Impact Statement

This paper aims to improve the reliability and efficiency of multimodal large language models by mitigating visual hallucinations without relying on external feedback during preference alignment. Such methods may benefit applications where faithful visual grounding is important, including assistive systems, scientific analysis, and decision-support tools. However, IRIS does not eliminate hallucinations entirely, and models aligned with our method should be carefully evaluated before deployment in real-world systems. It is not intended to replace human judgment in high-stakes applications.

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

# A. Algorithmic Details

---

**Algorithm 1** IRIS: Implicit Reward-Guided Internal Sifting (appendix pseudocode)

---

**Require:** $\mathcal{D}_{\text{SFT}} = \{(v, x, y^\star)\}$, base policy $\pi_{\theta_{\text{base}}}$, rounds $R$, candidates $K$, rectifier $\gamma$, loss weights $(\beta, \lambda, \delta)$
**Ensure:** aligned policy $\pi_{\theta_R}$

 1: **Warm-up (calibration).**
 2:     $\pi_{\theta_0} \leftarrow \text{SFT}(\pi_{\theta_{\text{base}}}, \mathcal{D}_{\text{SFT}})$
 3: **for** $r = 1$ **to** $R$ **do**
 4:     **(A) On-policy preference data construction.**
 5:     Scoring reference: $\pi_{\text{ref}}^{(r-1)} \leftarrow \begin{cases} \pi_{\theta_{\text{base}}}, & r = 1 \\ \pi_{\theta_{r-2}}, & r > 1 \end{cases}$
 6:     Initialize preference set $\mathcal{D}^{(r)} \leftarrow \emptyset$
 7:     **for** each $(v, x, y^\star) \in \mathcal{D}_{\text{SFT}}$ **do**
 8:       Sample $K$ candidates $\{y^{(k)}\}_{k=1}^K \sim \pi_{\theta_{r-1}}(\cdot \mid v, x)$
 9:       **for** $k = 1$ **to** $K$ **do**
10:         Compute implicit rewards $r_{\text{image}}^{(r)}(v, x, y^{(k)})$ and $r_{\text{text}}^{(r)}(x, y^{(k)})$     (Eqs. 8, 9)
11:         RVG score:
$$S^{(r)}(v, x, y^{(k)}) \leftarrow r_{\text{image}}^{(r)} - \gamma \max\left(0, \, r_{\text{text}}^{(r)} - r_{\text{image}}^{(r)}\right)$$
                                                                                               (Eq. 10)
12:       **end for**
13:       Select extrema: $y_w \leftarrow \arg\max_k S^{(r)}(v, x, y^{(k)})$, $y_l \leftarrow \arg\min_k S^{(r)}(v, x, y^{(k)})$     (Eq. 11)
14:       Filter low-confidence pairs and anchor with $y^\star$ if needed (Sec. 4)
15:       Add $(v, x, y_w, y_l)$ to $\mathcal{D}^{(r)}$
16:     **end for**
17:     **(B) Grounded preference learning.**
18:     Optimization reference: freeze $\pi_{\text{ref}} \leftarrow \pi_{\theta_{r-1}}$ and initialize $\pi_\theta \leftarrow \pi_{\theta_{r-1}}$
19:     For each $(v, x, y_w, y_l) \in \mathcal{D}^{(r)}$, form negative image $\tilde{v} \leftarrow T(v)$ (App. C)
20:     Update $\pi_\theta$ by minimizing $\mathcal{L}_{\text{total}} = \mathcal{L}_{\text{ctp}} + \lambda \mathcal{L}_{\text{cvp}} + \mathcal{L}_{\text{anchor}}$     (Eq. 15)
21:     Set $\pi_{\theta_r} \leftarrow \pi_\theta$
22: **end for**
23: **return** $\pi_{\theta_R}$

---

# B. Detailed Computational Cost Analysis

This section reports the wall-clock cost of `IRIS` and clarifies the primary sources of its efficiency. The key advantage of `IRIS` is its **lightweight sampling-and-sifting pipeline**. We construct preference pairs using only intrinsic log-likelihood signals from the policy, completely bypassing external evaluators or complex multi-stage verification.

**Context: external-feedback pipelines.** A major cost driver in external-feedback methods is the scoring stage. For instance, He et al. (2026) report that generating and scoring a 22k preference dataset for RLAIF-V takes approximately **66 hours** on 8×NVIDIA A100 GPUs. The bottleneck stems from RLAIF-V's "Divide-and-Conquer" strategy, which requires decomposing responses into multiple claims and conducting repeated QA-based inference with a large labeler model (e.g., 34B) to verify each claim. We include this figure as context, noting that while the dataset scale and hardware differ, it represents the typical overhead of prompt-based external feedback.

**IRIS data curation cost.** `IRIS` eliminates this dependency by computing scores directly in the model's native log-probability space during or immediately after the sampling process. The additional overhead is minimal, as it only requires evaluating log-probabilities for a small set of $K = 5$ candidates. On a single node with 8×NVIDIA H20 GPUs, curating our 5.7k on-policy dataset takes **1.5 hours** in total (**1.0** hour for on-policy sampling and **0.5** hour for implicit-reward sifting).

**Normalized view of curation cost.** To better reflect the pipeline-level difference, Table 7 reports the curation time normalized by dataset size (hours per 1k prompts). Despite using H20 GPUs, `IRIS` achieves a significantly lower

normalized cost (0.26h vs. 3.00h). This gap (11.5× in Time/1k) suggests that the dominant cost difference comes from the pipeline design—most notably, avoiding labeler-model inference—though the two numbers are measured under different hardware and implementations.

*Table 7.* Curation-time comparison (context vs. IRIS). "Time/1k" normalizes wall-clock time by dataset size ($T_{total}$/Size × 1000). Numbers for RLAIF-V are taken from He et al. (2026); IRIS times are measured in our implementation.

| Method | Dataset size | Total curation time (h) | Time/1k (h) |
|---|---|---|---|
| RLAIF-V (reported) | 22k | 66.0 | 3.00 |
| IRIS (ours) | 5.7k | 1.5 | 0.26 |

**End-to-end turnaround time.** The wall-clock time for one full IRIS round (generation → scoring → optimization) is:

- **On-policy sampling:** 1.0 hour (8×H20)
- **Implicit-reward sifting:** 0.5 hour (8×H20)
- **DPO training:** 1.0 hour (8×H20)

The entire cycle completes in **2.5 hours**. This rapid turnaround allows for efficient iterative alignment, a key feature that distinguishes our approach from more computationally intensive self-alignment frameworks.

## C. Additional Implementation Details

**Rejected image construction.** Following prior work (Fu et al., 2025; Liu et al., 2025), we construct rejected images $\tilde{v}$ by perturbing the original image $v$ to serve as negative visual inputs in Eq. 13. We consider four augmentation strategies: **Black** (all-zero image), **Random** (replace $v$ with a randomly sampled image from the training set), **Crop** (random crop followed by resizing back to the original resolution), and **Diffusion**. For diffusion, we apply the DDPM forward noising process with a total horizon of $T=1000$ and a fixed timestep $t=500$:

$$x_t = \sqrt{\bar{\alpha}_t}\, x_0 + \sqrt{1 - \bar{\alpha}_t}\, \epsilon, \; \epsilon \sim \mathcal{N}(0, I),$$

and use $x_t$ as $\tilde{v}$ (forward noising only; no denoising model is used). Table 8 reports an ablation on AMBER; all results are based on the same base model (LLaVA-1.5-7B), where R1/R2 denote the first/second IRIS preference round.

*Table 8.* Ablation on data augmentation strategies for constructing negative samples.

| MEASURE | ROUND | CHAIR↓ | AMBER HALRATE↓ | COG↓ |
|---|---|---|---|---|
| BLACK | R1 | 4.1 | 19.9 | 1.9 |
| BLACK | R2 | 3.9 | 19.3 | 1.9 |
| RANDOM | R1 | 4.1 | 20.4 | 1.9 |
| RANDOM | R2 | 4.4 | 21.6 | 1.9 |
| CROP | R1 | 3.8 | 18.6 | 1.5 |
| CROP | R2 | 3.2 | 15.7 | 1.0 |
| DIFFUSION | R1 | 3.8 | 17.5 | 1.6 |
| DIFFUSION | R2 | **2.4** | **11.3** | **1.1** |

**Pair screening, length-aware filtering, and conflict anchoring.** To improve the quality of on-policy preference supervision, we apply a lightweight post-processing pipeline before optimization. We first score $K$ sampled candidates per prompt and form a raw preference pair by selecting the highest- and lowest-scoring candidates. We then perform screening to remove unreliable or degenerate pairs, and apply a length-aware filter on descriptive prompts to reduce length bias. Pairs flagged by the length filter are restored by replacing the preferred side with the corresponding SFT reference from $\mathcal{D}_{SFT}$. Finally, for pairs whose preference direction clearly conflicts with the SFT reference, we conservatively anchor them to the SFT ordering. All steps are internal to the training pipeline and require no external evaluator.

---

**Algorithm 2** Post-processing for on-policy preference pairs

---

**Require:** Prompts $(v, x)$, SFT references $\mathcal{D}_{\text{SFT}}$, repeat $K$, scoring function $r(v, x, y)$
**Ensure:** Final preference pairs $\mathcal{P}$
1: $\mathcal{P} \leftarrow \emptyset$
2: **for** each prompt $(v, x)$ **do**
3:    Sample $K$ candidates $\{y_j\}_{j=1}^K \sim \pi_\theta(\cdot \mid v, x)$ and compute scores $r_j = r(v, x, y_j)$
4:    $y_w \leftarrow \arg\max_j r_j, \quad y_l \leftarrow \arg\min_j r_j$
5:    $y_{\text{sft}} \leftarrow \mathcal{D}_{\text{SFT}}(v, x)$ {if available}
   {(i) Screening: remove unreliable/degenerate pairs}
6:    **if** $\text{norm}(y_w) = \text{norm}(y_l)$ **or** $\text{invalid}(y_w, y_l)$ **then**
7:       **continue**
8:    **end if**
   {(ii) Length-aware filtering: reduce length bias on descriptive prompts}
9:    **if** $\text{LenFilter}(v, x, y_w, y_l)$ and $y_{\text{sft}}$ exists **then**
10:      $y_w \leftarrow y_{\text{sft}}$ {restore preferred side}
11:    **end if**
   {(iii) Conflict anchoring: enforce preference direction consistency}
12:    **if** $y_{\text{sft}}$ exists and $\text{Conflict}(y_w, y_l, y_{\text{sft}})$ **then**
13:      $y_w \leftarrow y_{\text{sft}}$
14:    **end if**
15:    $\mathcal{P} \leftarrow \mathcal{P} \cup \{(v, x, y_w, y_l)\}$
16: **end for**
17: **Return** $\mathcal{P}$

---

## D. Impact on General Capabilities and Training Dynamics

**Effect of anchored regularization on general capability.** Table 9 reports general instruction-following performance on LLaVA-Bench (in-the-Wild). Compared to the base model, removing the anchored regularization leads to a noticeable drop in overall accuracy, indicating degraded general capability, even though hallucination-related metrics improve. This behavior is expected for preference-based optimization, which primarily enforces relative ranking between responses and may reduce the absolute likelihood of preferred outputs. The anchored regularization mitigates this issue by constraining the reference-relative reward of preferred responses, thereby stabilizing training and preserving broad instruction-following ability while optimizing for hallucination mitigation.

*Table 9.* **Effect of anchored regularization on general capability on LLaVA-Bench (in-the-Wild) (Accuracy %).** All results are evaluated after Round 2. Higher is better.

| METHOD | OVERALL↑ | COMPLEX↑ | CONV↑ | DETAIL↑ |
|---|---|---|---|---|
| BASE MODEL | 55.7 | **64.8** | 50.3 | 46.2 |
| W/O ANCHOR | 52.3 | 48.5 | **62.2** | 47.2 |
| OURS (FULL) | **56.4** | 57.5 | 60.6 | **49.2** |

*Table 10.* **LLaVA-1.5-7B: LLaVA-Bench (in-the-Wild) accuracy across rounds (Accuracy %).** Higher is better.

| STAGE | OVERALL↑ | COMPLEX↑ | CONV↑ | DETAIL↑ |
|---|---|---|---|---|
| BASE MODEL | 55.7 | 64.8 | 50.3 | 46.2 |
| ROUND 1 | 56.3 | 56.8 | 60.4 | 50.4 |
| ROUND 2 | 56.4 | 57.5 | 60.6 | 49.2 |
| ROUND 3 | 53.2 | 54.5 | 57.6 | 45.9 |

## E. External Discrete Scoring Analysis

We conduct a diagnostic analysis of external discrete scoring on self-generated candidates. Using Qwen3-VL-30B-A3B-Instruct to score 28,665 candidate responses generated from 5,733 prompts at the R0 stage, we find that a 1–3 scale yields a 94.47% pairwise tie rate and a 15.16% all-tie rate; even with a 1–5 scale, the pairwise tie rate remains 86.27%. We further compare external scores with the final IRIS preference pairs and find that 2,339 pairs (40.8%) receive identical

*Table 11.* **LLaVA-1.5-13B: LLaVA-Bench (in-the-Wild) accuracy across rounds (Accuracy %).** Higher is better.

| STAGE | OVERALL↑ | COMPLEX↑ | CONV↑ | DETAIL↑ |
|---|---|---|---|---|
| BASE MODEL | 64.9 | 72.1 | 64.5 | 52.8 |
| ROUND 1 | 64.8 | 67.2 | 65.8 | 59.1 |
| ROUND 2 | 65.9 | 66.2 | 74.1 | 55.8 |
| ROUND 3 | 61.8 | 64.6 | 63.4 | 55.0 |

*Figure 5.* **Example of qualitative analysis for Round 3.** An example of the model's performance at the 3rd iteration in reducing descriptive illusions and maintaining visual consistency.

external scores for the chosen and rejected responses. On these externally tied pairs, the internal RVG score still shows a continuous gap, with mean $\Delta S = 0.0107$ and standard deviation 0.0056. These results suggest that implicit rewards provide finer-grained preference information than external discrete scoring for self-generated candidates.

# F. Cross-Architecture Evaluation

To further examine whether IRIS can provide useful grounding signals beyond the LLaVA family, we conduct a preliminary one-round experiment on Qwen2.5-VL-3B. Due to computational constraints, this experiment is not intended as a full multi-round IRIS evaluation, but rather as an initial cross-architecture validation. The IRIS sampling-and-sifting pipeline is applied without using external feedback during preference alignment.

As shown in Table 12, IRIS consistently reduces hallucination-related metrics on both AMBER and Object HalBench, while largely maintaining comparable object coverage. These results provide preliminary evidence that the proposed implicit-reward-based preference construction may not be limited to the LLaVA-1.5 architecture.

*Table 12.* Preliminary one-round evaluation of IRIS on Qwen2.5-VL-3B. Best results are in **bold**.

| Model | AMBER | | | | Object Hal | |
|---|---|---|---|---|---|---|
| | CHAIR↓ | Cover↑ | HalRate↓ | Cog↓ | CHAIRs↓ | CHAIRi↓ |
| Qwen2.5-VL-3B | 6.7 | **64.0** | 37.9 | 2.8 | 16.3 | 9.7 |
| +IRIS-R1 | **3.6** | 63.2 | **22.7** | **1.5** | **11.8** | **6.0** |

# G. Hyperparameter Sensitivity Analysis

*Table 13.* Ablation study on $\lambda$. We focus on hallucination-related metrics. **Bold** indicates the best performance.

| SETTING | AMBER | | | OBJECT HAL | |
|---|---|---|---|---|---|
| | CHAIR↓ | HALRATE↓ | COG↓ | CHAIRs↓ | CHAIRI↓ |
| $\lambda = 0$ | 4.9 | 25.4 | 2.2 | 17.0 | 7.98 |
| $\lambda = 0.3$ | 3.8 | 19.7 | 2.0 | 18.6 | 9.41 |
| $\lambda = 0.5$ | 3.2 | 15.8 | 1.5 | 15.3 | 8.00 |
| $\lambda = 0.7$ | 3.0 | 14.4 | 1.4 | 11.6 | 5.92 |
| $\lambda = 1.0$ | **2.4** | **11.3** | **1.1** | **8.6** | **4.56** |
| $\lambda = 1.2$ | 2.8 | 12.9 | 1.4 | 9.6 | 5.02 |

*Table 14.* Sensitivity analysis of the rectification strength $\gamma$ in RVG. Results are reported for Round 1 and Round 2. Lower is better.

| VALUE | ROUND | CHAIR↓ | COG↓ | HALRATE↓ |
|---|---|---|---|---|
| 0.0 | R1 | 3.8 | 1.8 | 17.8 |
| 0.0 | R2 | 2.8 | 1.2 | 12.8 |
| 0.1 | R1 | 4.1 | 1.7 | 18.5 |
| 0.1 | R2 | 2.7 | 1.4 | 12.5 |
| 0.3 | R1 | 4.1 | 1.8 | 19.1 |
| 0.3 | R2 | 2.7 | 1.4 | 13.0 |
| 0.5 | R1 | 4.5 | 2.0 | 20.9 |
| 0.5 | R2 | 2.7 | 1.3 | 12.7 |
| 0.7 | R1 | 3.8 | 1.6 | 17.5 |
| 0.7 | R2 | **2.4** | **1.1** | **11.3** |
| 1.0 | R1 | 4.1 | 1.8 | 19.4 |
| 1.0 | R2 | 3.0 | 1.6 | 14.4 |
| 1.5 | R1 | 4.3 | 1.8 | 20.3 |
| 1.5 | R2 | 3.2 | 1.6 | 15.7 |
| 5.0 | R1 | 4.6 | 2.1 | 22.5 |
| 5.0 | R2 | 2.9 | 1.4 | 14.2 |
| 20.0 | R1 | 4.5 | 2.1 | 21.8 |
| 20.0 | R2 | 3.2 | 1.6 | 16.8 |

# H. Theoretical Analysis: Learning from Self-Generated Preferences

This appendix provides a theoretical analysis of why IRIS can learn from noisy self-generated preference pairs. We (i) derive a gradient difference form for a standard pairwise loss, (ii) show that each gradient step locally increases the log-likelihood margin on the constructed pair, and (iii) argue that selecting the best and worst among $K$ candidates enlarges the expected true-quality gap, which strengthens the delta-learning premise (Geng et al., 2025).

## H.1. Setup and Notation

Let $c = (v, x)$ denote the multimodal context, and let $\pi_\theta(y \mid c)$ be the policy. Assume an unobserved grounding quality function $s^*(c, y) \in \mathbb{R}$. Given $K$ candidates $\mathcal{Y}_K = \{y^{(1)}, \ldots, y^{(K)}\} \sim \pi_\theta(\cdot \mid c)$, IRIS constructs a preference pair $(y_w, y_l)$ by selecting a high-score response as the winner and a low-score response as the loser (using the scoring rule in the main text).

## H.2. Pairwise Gradient Difference Form

**Lemma H.1** (Pairwise Gradient Difference Form). *Consider the pairwise preference loss*

$$\mathcal{L}_{pref}(c, y_w, y_l) = -\log \sigma\big(\Delta_\theta(c, y_w, y_l)\big), \tag{16}$$

*where*

$$\Delta_\theta(c, y_w, y_l) = \beta\left(\log\frac{\pi_\theta(y_w \mid c)}{\pi_{ref}(y_w \mid c)} - \log\frac{\pi_\theta(y_l \mid c)}{\pi_{ref}(y_l \mid c)}\right), \tag{17}$$

*with $\beta > 0$ and a fixed reference policy $\pi_{ref}$. Then the gradient satisfies*

$$\nabla_\theta \mathcal{L}_{pref}(c, y_w, y_l) = -w_\theta(c, y_w, y_l)\Big(\nabla_\theta \log \pi_\theta(y_w \mid c) - \nabla_\theta \log \pi_\theta(y_l \mid c)\Big), \tag{18}$$

*where $w_\theta(c, y_w, y_l) = \beta\,\sigma\big(-\Delta_\theta(c, y_w, y_l)\big) \in (0, \beta)$.*

*Proof.* By the chain rule, $\nabla_\theta \mathcal{L}_{pref} = -\sigma(-\Delta_\theta)\nabla_\theta \Delta_\theta$. Since $\pi_{ref}$ is fixed, $\nabla_\theta \Delta_\theta = \beta\big(\nabla_\theta \log \pi_\theta(y_w \mid c) - \nabla_\theta \log \pi_\theta(y_l \mid c)\big)$, which yields the claim. □

## H.3. Margin Improvement and a Delta-Learning Premise

**Assumption H.2** (Average Directional Correctness). The constructed preference pairs satisfy a positive expected quality gap:

$$\mathbb{E}\big[s^*(c, y_w) - s^*(c, y_l)\big] \geq \delta, \qquad \delta > 0, \tag{19}$$

where the expectation is over contexts and the randomness in sampling and pair construction.

**Proposition H.3** (Local Margin Improvement). *Define the log-likelihood margin of the constructed pair as*

$$m_\theta(c) = \log \pi_\theta(y_w \mid c) - \log \pi_\theta(y_l \mid c). \tag{20}$$

*For the update $\theta' = \theta - \eta\nabla_\theta \mathcal{L}_{pref}$ with sufficiently small $\eta > 0$,*

$$m_{\theta'}(c) = m_\theta(c) + \eta\,w_\theta(c, y_w, y_l)\left\|\nabla_\theta m_\theta(c)\right\|^2 + o(\eta), \tag{21}$$

*and therefore $m_{\theta'}(c) \geq m_\theta(c)$ whenever $\nabla_\theta m_\theta(c) \neq 0$.*

*Proof.* From Lemma H.1, $-\nabla_\theta \mathcal{L}_{pref} = w_\theta(c, y_w, y_l)\nabla_\theta m_\theta(c)$. A first-order Taylor expansion gives

$$\begin{aligned}
m_{\theta'}(c) &= m_\theta(c) + \eta\,\langle\nabla_\theta m_\theta(c),\, w_\theta(c, y_w, y_l)\nabla_\theta m_\theta(c)\rangle + o(\eta) \\
&= m_\theta(c) + \eta\,w_\theta(c, y_w, y_l)\left\|\nabla_\theta m_\theta(c)\right\|^2 + o(\eta),
\end{aligned} \tag{22}$$

which yields the result. □

**Interpretation: Focusing on Violated Preferences.** Proposition H.3 shows that a gradient step locally increases the log-likelihood margin on the constructed pair. The weight $w_\theta = \beta\sigma(-\Delta_\theta)$ emphasizes *violated* preferences: when $\Delta_\theta < 0$, the model ranks the loser above the winner under the implicit margin, and $w_\theta$ becomes large; when $\Delta_\theta > 0$, the preference is already satisfied and $w_\theta$ becomes small. Combined with Assumption H.2, this implies that training concentrates updates on informative disagreements, while occasional construction errors do not dominate in expectation, consistent with the delta learning view (Geng et al., 2025).

## H.4. Signal Amplification via Best-of-$K$ Sifting

**Proposition H.4** (Extrema Selection Amplifies the Expected Quality Gap). *Let $(y_w^{(K)}, y_l^{(K)})$ be the winner/loser obtained by selecting the maximum/minimum score from $K$ i.i.d. samples $\mathcal{Y}_K \sim \pi_\theta(\cdot \mid c)$. Assume the score is positively related to $s^*(c, y)$ in the sense that higher-score selections tend to have higher expected $s^*$. Then the expected quality gap is non-decreasing in $K$:*

$$\mathbb{E}\big[s^*(c, y_w^{(K)}) - s^*(c, y_l^{(K)})\big] \geq \mathbb{E}\big[s^*(c, y_w^{(2)}) - s^*(c, y_l^{(2)})\big]. \tag{23}$$

*Proof sketch.* Let $U_i = s^*(c, y^{(i)})$ be i.i.d. draws. In the ideal case where the score preserves the ordering of $U_i$, the selected pair corresponds to $(\max_i U_i, \min_i U_i)$, and the expected range $\mathbb{E}[\max_i U_i - \min_i U_i]$ increases with $K$ by standard order-statistics. With imperfect but positive relation, selecting extrema by the score still tends to choose a winner with larger $U$ and a loser with smaller $U$ than a random pair, preserving the monotonic trend in expectation. □

# I. Qualitative Examples of Model Response

To provide an intuitive understanding of the IRIS framework's efficacy, we present qualitative examples from our evaluation benchmarks. These instances illustrate the trajectory of model improvement throughout the **iterative refinement rounds**, highlighting how the final model (**R2**) successfully rectifies hallucinations observed in baselines or earlier iterations. In the provided examples (e.g., Figure 6), red text denotes hallucinations or factual errors, while green text indicates factually grounded statements.

## I.1. Visualization of the Preference Refinement Process

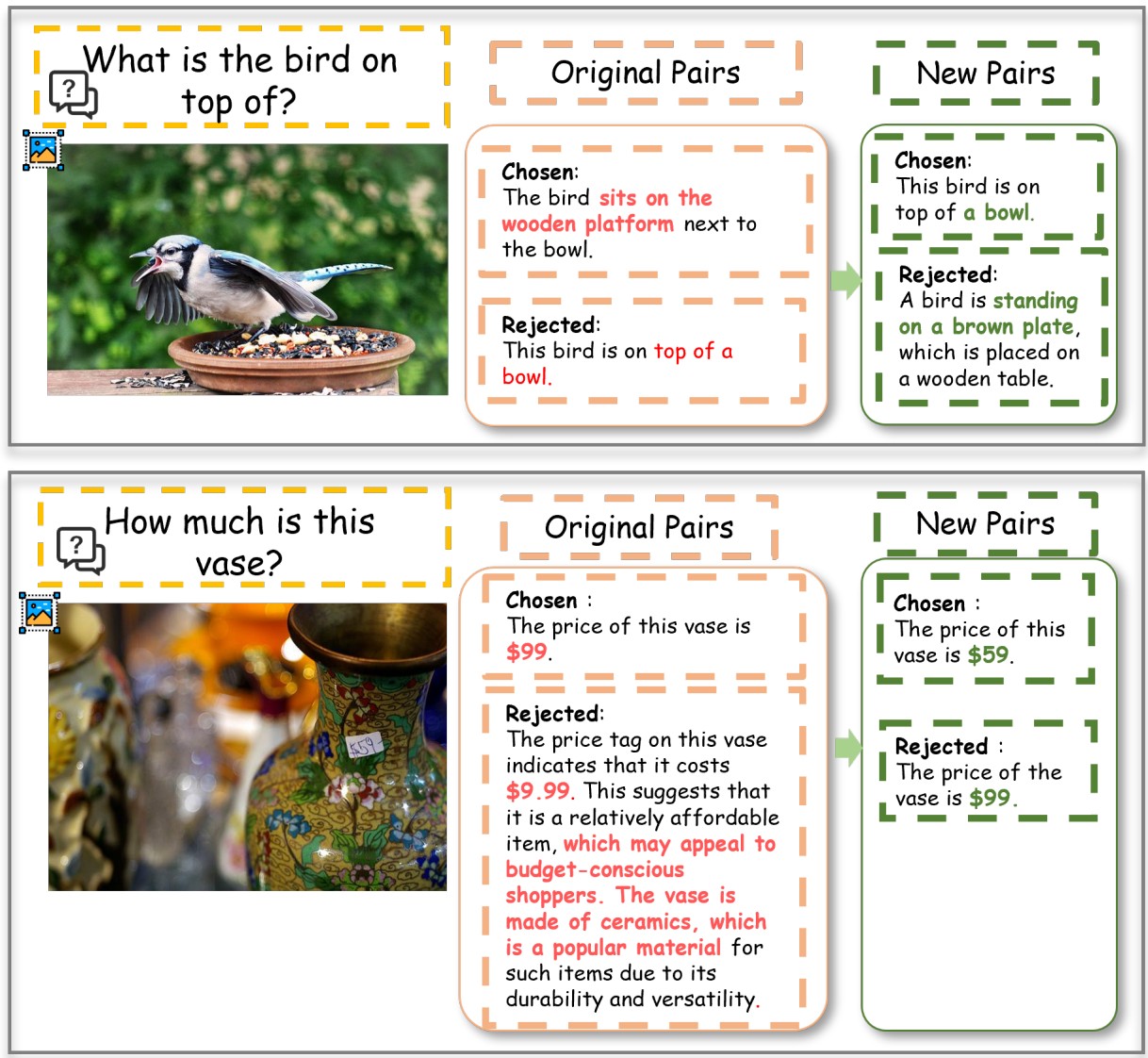

*Figure 6.* **Preference Pair Refinement (VQA Task).** Illustration of how IRIS sifts and refines preference pairs to mitigate object hallucination (e.g., bird location, vase price).

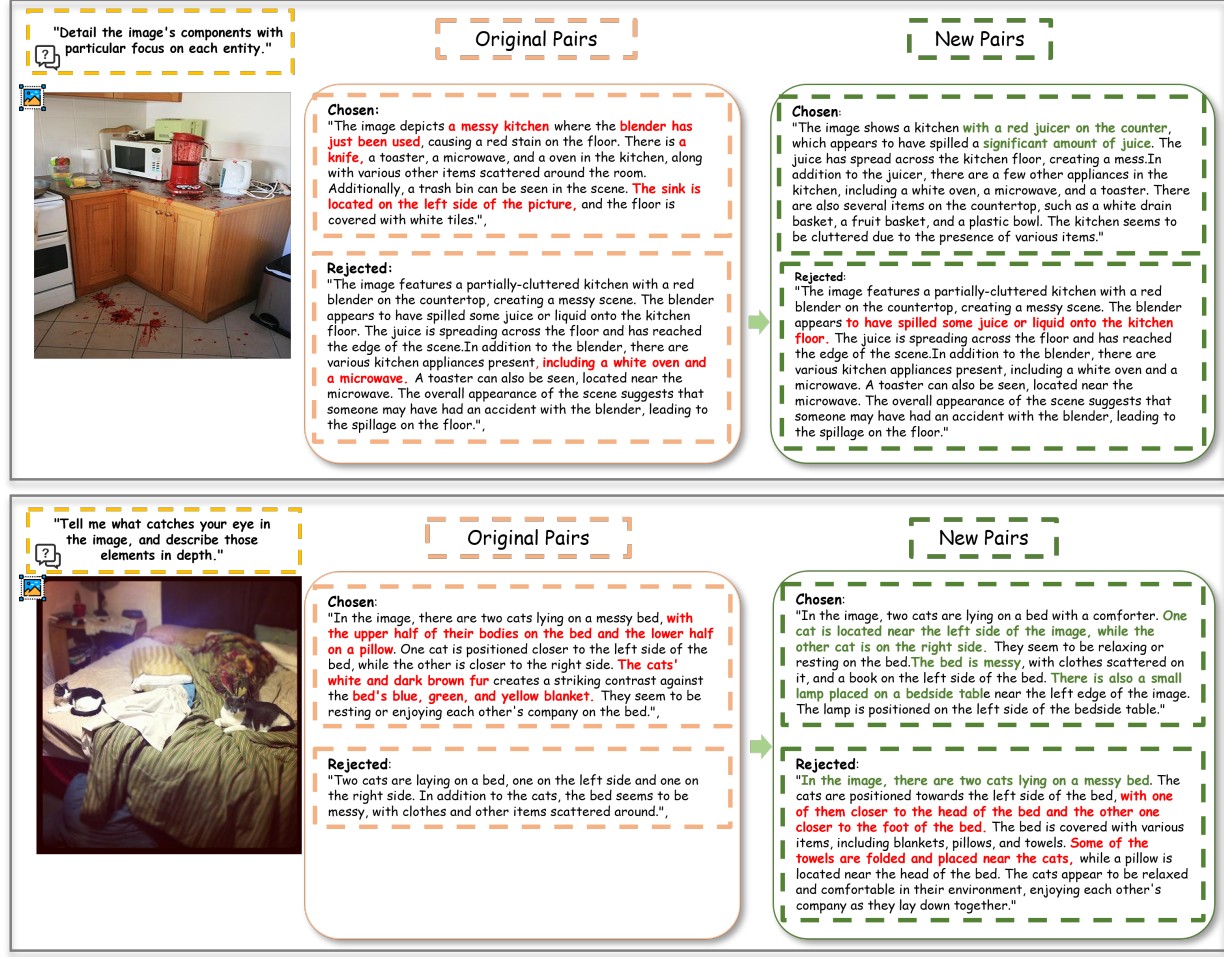

*Figure 7.* **Preference Pair Refinement (Description Task).** Demonstration of preference evolution for detailed image descriptions. The model learns to reject detailed but hallucinated descriptions in favor of visually grounded ones.

## I.2. Qualitative Comparisons on Detail Description

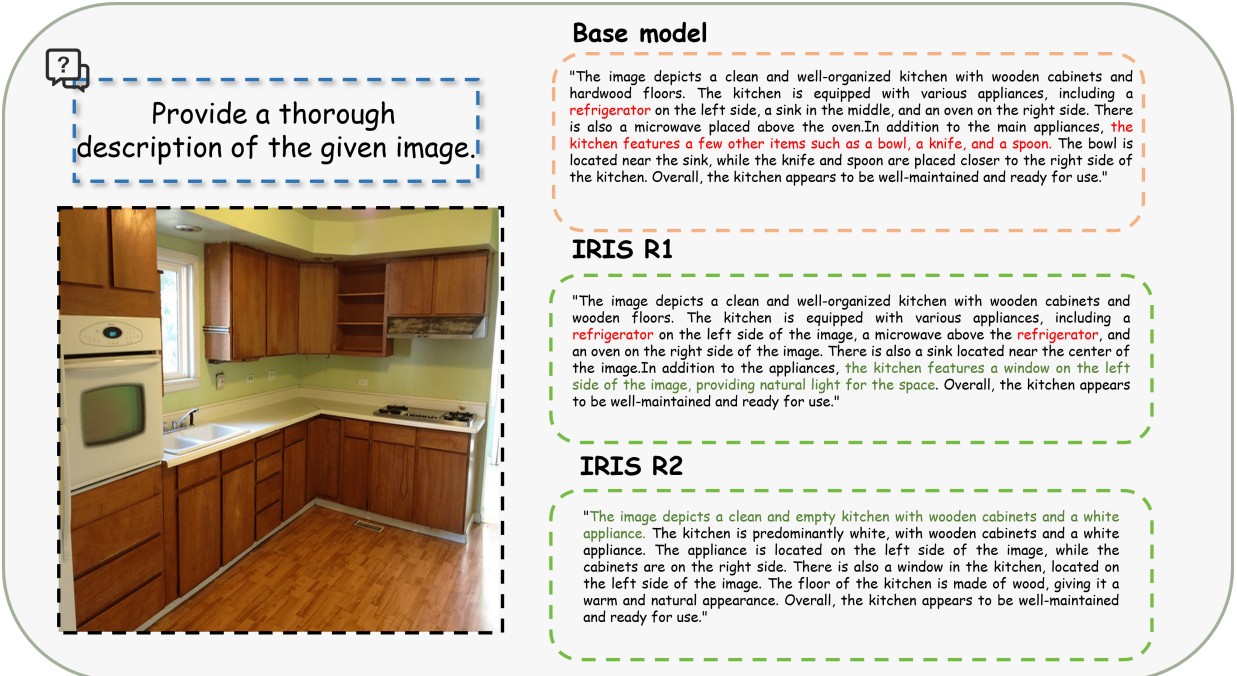

*Figure 8.* **Mitigating Descriptive Hallucination.** The initial base model hallucinates non-existent objects such as a bowl, knife, and spoon. As the training progresses, IRIS(R2) correctly identifies the empty counter and wooden cabinets, adhering strictly to visual evidence.

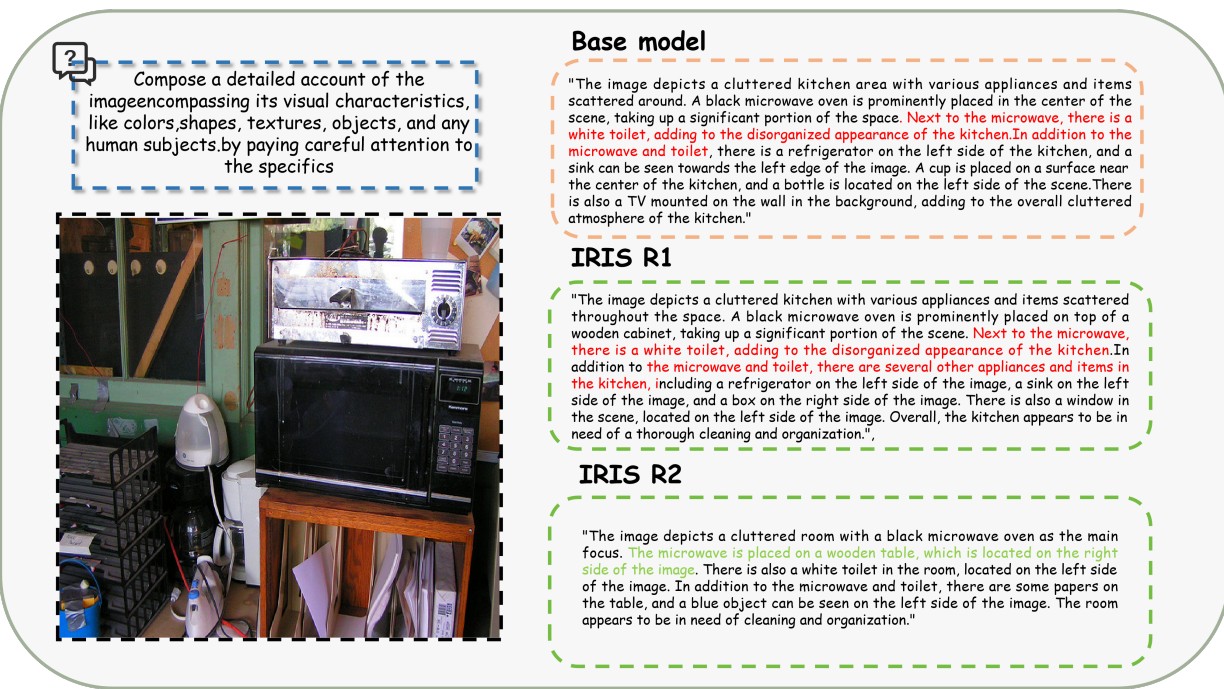

*Figure 9.* **Mitigation of Scene Completion Hallucination.** Unlike the base model, which erroneously infers a TV on the wall and a bottle on the left, IRIS(R2) avoids fabricating unseen background elements in the cluttered scene.

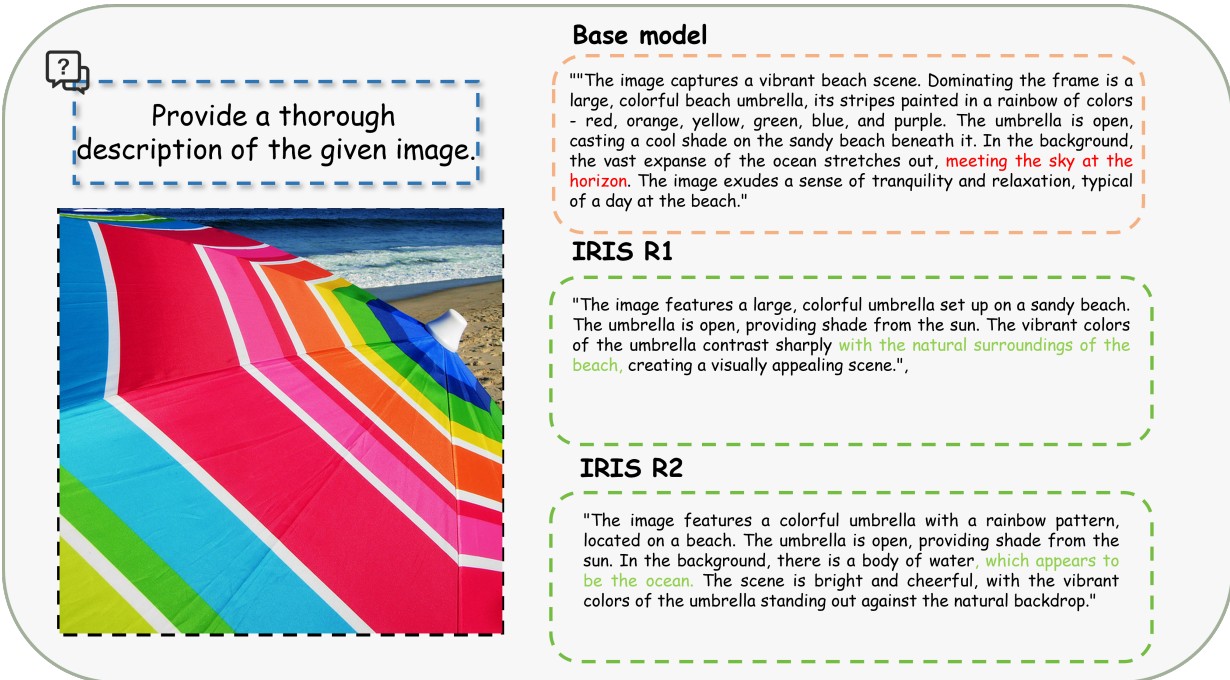

*Figure 10.* **Misidentification Correction in Complex Scenarios.** Illustrates how the model reduces incorrect identifications (e.g., mistaking the ocean/sky boundary) and unnecessary inferences in multi-object scenarios.

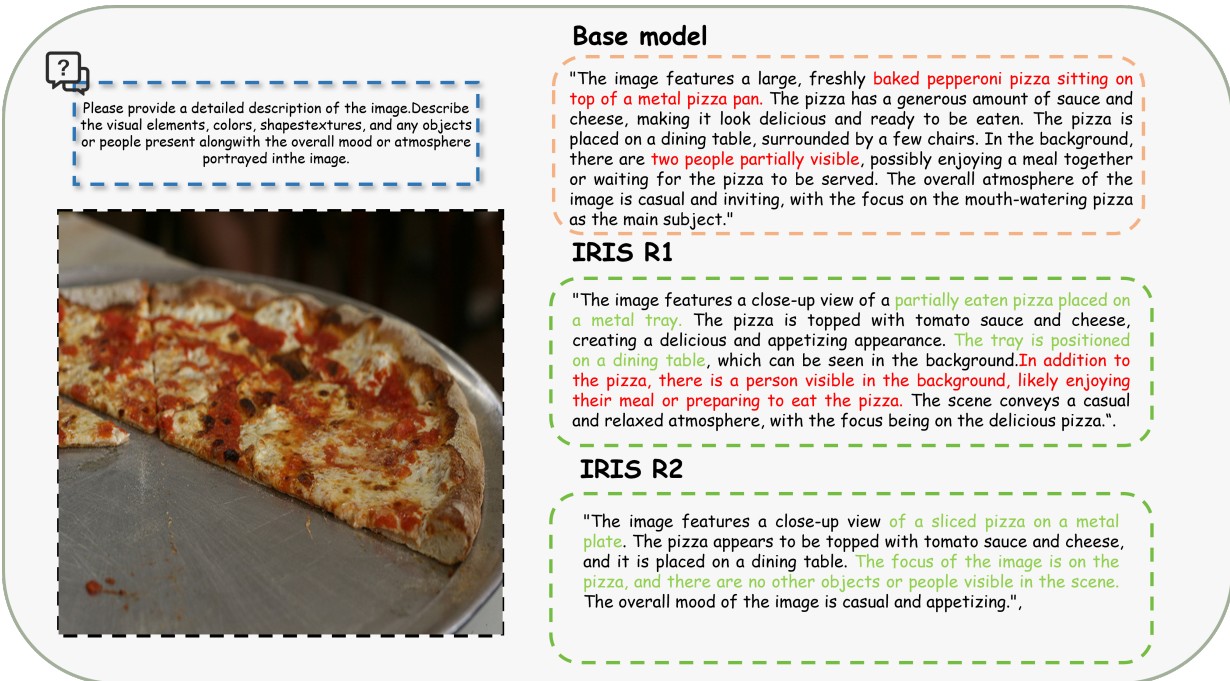

*Figure 11.* **Detail Preservation and Negative Responses.** This demonstrates the model's ability to choose conservative (non-false) responses when faced with uncertain details, correcting the hallucination of "two people partially visible" in the background.

