# OpenReview forum: "IRIS: Implicit Reward-Guided Internal Sifting for Mitigating Multimodal Hallucination"
_ICML.cc/2026/Conference — ICML 2026 regular_

### Official Review · Reviewer_Y5NW · 2026-03-09

**Soundness:** 2
**Presentation:** 2
**Significance:** 2
**Originality:** 2
**Overall Recommendation:** 2
**Confidence:** 5

**Summary:**

This paper targets hallucination mitigation in multimodal large language models via preference optimization. It proposes IRIS, performs an SFT warm-up followed by DPO, aiming to avoid reliance on human-labeled preferences or external evaluators during preference construction. For each (image, text) input, the method samples multiple candidate responses from the current policy and scores them using a self-derived “implicit reward” (a log-likelihood ratio between the current policy and a reference policy). The highest- and lowest-scoring candidates are selected to form a preference pair for DPO training.
IRIS further introduces Rectified Visual Guidance (RVG), which compares image-conditioned and text-only (vision-removed) implicit rewards and down-weights candidates whose text-only score exceeds their image-conditioned score, with the intent of reducing responses driven primarily by language priors rather than visual evidence. The overall objective combines standard DPO with additional terms (including a visually perturbed preference component and an anchoring regularizer) to improve training stability. The paper reports improvements on several hallucination-related benchmarks relative to the chosen baselines.

**Compliance With Llm Reviewing Policy:**

Affirmed.

**Final Justification:**

1. I do not see any experimental results on base models beyond LLaVA.
2. Reporting preference-pair accuracy through manual inspection is not acceptable to me.
3. I still believe that, when preference data already uesd in your method, generating new pairs is unnecessary.

**Key Questions For Authors:**

1. **Model-family generalization:** Please report cross-family results (e.g., Qwen-VL, DeepSeek-VL, and other open-source MLLMs). Evaluating only on a single model family is not sufficient to support the claimed generality.

2. **Core novelty (scoring / preference construction):** Since your scoring mechanism is the main contribution, can you compare the preference pairs produced by your method against *fully correct* human-annotated preference pairs?
   - How do you quantify the quality of your scoring (e.g., pairwise direction accuracy, correlation with human judgments, gap in grounding quality)?
   - Please provide an additional experiment that directly validates the correctness of the scoring mechanism (not only downstream benchmark gains).

3. Your pipeline still relies on labeled data for the SFT warm-up, so the human-annotation cost is not eliminated. If the DPO stage avoids human labels but uses noisy self-generated pairs, why is this preferable to simply collecting (or using) human-annotated preferences for DPO?
   - Can you justify why noisy self-generated preference data should be *better* than human-annotated data?
   - The current theory argues learnability from noisy pairs, but the key missing point is whether your generated preferences are *superior to human-labeled preferences*; without that, the theoretical argument is less meaningful.

**Limitations:**

yes

**Strengths And Weaknesses:**

**Strengths**

1.Well-written and nicely formatted. The paper is visually clean and easy to follow, which improves readability.

2.The paper claims that, during the DPO stage, a human-annotation-free approach can outperform human-annotated supervision, which is a very bold claim.

**Weaknesses**

1.Model-family generalization: Experiments only on LLaVA families.


2.Low paradigm novelty / “stitching” concern. The overall recipe (SFT warm-up → DPO) is not new and has already attracted significant attention (e.g., OPA-DPO, CVPR 2025 Oral). The proposed Anchored Regularization and Conditional Visual Preference look highly similar to prior designs in mDPO. The broader idea of self-evolving/self-rewarding data has also appeared earlier (e.g., CSR, NeurIPS 2024). The clear novelty is preference-pair construction;

References:

[1]Wang F, Zhou W, Huang J Y, et al. mdpo: Conditional preference optimization for multimodal large language models[C]//Proceedings of the 2024 Conference on Empirical Methods in Natural Language Processing. 2024: 8078-8088.（mDPO）

[2]Yang Z, Luo X, Han D, et al. Mitigating hallucinations in large vision-language models via dpo: On-policy data hold the key[C]//Proceedings of the IEEE/CVF Conference on Computer Vision and Pattern Recognition. 2025: 10610-10620. （OPA-DPO）

[3]Zhou Y, Fan Z, Cheng D, et al. Calibrated self-rewarding vision language models[J]. Advances in Neural Information Processing Systems, 2024, 37: 51503-51531.（CSR）


3.**Automatic preference construction:** From the formulation, $r_{image}$ is the dominant signal and largely increases the relative probability of the current model’s own outputs, which risks self-reinforcement. This creates a potential mismatch between the optimization target (amplifying model-preferred outputs) and the intended goal (improving visual grounding / reducing hallucinations).

---

> ### Author Rebuttal · Authors · 2026-03-31
>
> Dear Reviewer Y5NW,
>
> We sincerely thank you for the constructive feedback and for recognizing the clarity of our presentation and the ambition of our human-annotation-free DPO claim.
>
> ---
>
> **Response to Weakness 1 and Question 1:**
>
> To demonstrate cross-architecture generalization, we additionaly evaluated IRIS on **Qwen2.5-VL-3B**. The results confirm its effectiveness beyond the LLaVA family and will be added to the revised manuscript.
>
> ---
>
> **Response to Weakness 2:**
>
> Thanks for outlining these prior methods; however, IRIS is totally different compared with them. While the overall SFT warm-up + DPO pipeline is not new, this is not the contribution we claim. We also do not present the loss form itself as a primary novelty, since this type of DPO-style loss was already introduced in mDPO and is now commonly used in later multimodal alignment methods, including OPA-DPO.
> The main novelty of IRIS lies in its grounding-aware on-policy preference construction. In OPA-DPO, SFT is mainly used to make external expert corrections learnable under DPO. In contrast, in IRIS, SFT warm-up serves a different purpose: it calibrates the initial implicit reward landscape for later internal scoring and sifting.
>
> IRIS also differs from methods such as CSR, which rely on external scoring signals. Instead, IRIS works directly in the model's native log-probability space. In particular, RVG compares the image-conditioned and text-only implicit rewards for the same response, and downweights responses supported more by language priors than by visual evidence. We will also clarify our contributions more clearly in the revised version.
>
> ---
>
> **Response to Weakness 3:**
>
> **Thanks for** outlining **this. However**, **we would like to clarify that IRIS does not reduce to generic self-reinforcement due to the innovative method of data construction.** IRIS does not construct pairs from **`r_image`** alone. Instead, **RVG** compares the **image-conditioned** and **text-only** implicit rewards for the same response, and penalizes cases where **`r_text > r_image`**, which indicates stronger reliance on **language priors** than on **visual evidence**. As a result, the resulting score is therefore **grounding-aware**, not a raw self-preference score.
>
> This is also consistent with **Table 4**. If IRIS mainly amplified already-preferred outputs, then **`r_image`** alone should already be sufficient. However, **RVG** performs better than **`r_image`** alone, showing that the gain comes from suppressing **language-prior-driven responses**, rather than **generic self-reinforcement**.
>
> ---
>
> **Response to Question 2:**
>
> We thank the reviewer for this suggestion. Since the **scoring and preference construction mechanism** is the core contribution of IRIS, we agree it should be validated **directly**, beyond downstream benchmark gains.
>
> To this end, we add a **pair-quality analysis** on **300 randomly sampled preference pairs** constructed by IRIS. For each pair, we manually inspect the **image**, **question**, and **chosen/rejected responses**, and judge whether the **chosen** response is better grounded. We find that the **chosen** response is better in **69%** of cases, the **rejected** response is better in **13%**, and the remaining **18%** are **ambiguous even under manual inspection**.
>
> This provides direct evidence that the induced preference direction is **correct in a clear majority of cases**, while most remaining cases are **hard or ambiguous**, rather than systematic reversals. We will include this analysis in the revised paper to directly validate the proposed scoring mechanism.
>
> ---
>
> **Response to Question 3:**
>
> We agree that IRIS does not eliminate human annotation cost in the full pipeline, since the **SFT warm-up uses labeled data**. Our claim is narrower: IRIS removes the need for **human-annotated preferences or external evaluators during the DPO stage**.
>
> We also do not claim that self-generated preferences are universally better than human annotations. Our point is more specific: under **iterative KL-constrained DPO**, **on-policy self-generated pairs** are better matched to the model’s current support, and are therefore more effective for iterative refinement. This is supported by **Table 3**, where replacing **static human pairs** with **self-generated pairs** leads to faster and more sustained gains, while human pairs plateau earlier.
>
> Our theory is also limited in scope. It shows that learning can progress from **noisy self-generated pairs** when the preference direction is correct on average, and that **best-vs-worst sifting** increases the expected gap. It does not claim superiority over human-labeled data.
>
> Practically, the advantage is therefore not only reduced reliance on external feedback, but also **scalability and turnaround**, since IRIS constructs preference pairs directly from intrinsic signals without external evaluators.

---

> > ### Author Rebuttal · Reviewer_Y5NW · 2026-04-06
> >
> > 1.
> > a. **A much larger portion of the experiments should be conducted on newer models.**
> > The main experiments should be updated to 7B or larger modern VLMs rather than relying primarily on LLaVA-1.5. Hallucination patterns in models from two years ago are not necessarily representative of current models, so evidence on outdated backbones is no longer sufficient. A much larger portion of the experiments should be conducted on newer models.
> >
> > b. **I do not see any actual results pasted in the rebuttal.**
> >
> > 2. **What is the standard for your manual inspection, and how is reproducibility ensured?**This is very hard for me to accept. Without a clearly defined protocol and evaluation standard, I do not see how the results can be reproduced or kept consistent across annotators.
> >
> >
> > 3.
> > I have read OPA-DPO very carefully. It was published more than a year ago, yet I still do not see a clear advantage of your method over it.
> >
> >
> > **My point is this:**
> > Since your method already relies on human-labeled data, I do not see why the DPO stage avoids using it and instead generates new preference pairs. To me, this feels somewhat unnecessary.
> >
> > **Reason: **
> > after SFT on RLHF-V, the model’s policy has already been shifted and its support expanded.
> > In that sense, the labeled data are no longer clearly off-policy or out-of-distribution, and the on/off-policy gap has already been largely mitigated. If so, I do not see why it is necessary to replace them with newly generated preference pairs that are much noisier and harder to control.
> > Moreover, I do not see how you can validate the quality of such data at scale. As you mentioned, the current approach relies on manual inspection, but then how can this be scaled up, and how can the quality be consistently guaranteed?
> >
> >
> >
> > I am sorry, but I will lower my score for three reasons:
> > 1. I do not see any experimental results on base models beyond LLaVA.
> > 2. Reporting preference-pair accuracy through manual inspection is not acceptable to me.
> > 3. I still believe that, when preference data already uesd in your method, generating new pairs is unnecessary.

---

> > > ### Author Response · Authors · 2026-04-08
> > >
> > > Dear Reviewer Y5NW,
> > > We thank you for the thoughtful feedback. Our responses to your concerns are as follows:
> > >
> > > > Architectural Generalization on Modern MLLMs
> > >
> > > We apologize that the additional experiments were not shown before. We have already evaluated IRIS on the newest Qwen2.5-VL-3B model during the rebuttal phase. Please refer to the anonymized link : [Anonymized Repository - Anonymous GitHub](https://anonymous.4open.science/r/code-D288/table/Effect.png)
> > >
> > > > Manual Evaluation Standard
> > >
> > > Thank you for raising this question. The manual inspection was conducted specifically in response to your previous question (Q.2) about whether our generated pairs align with human judgments. The evaluation protocol **followed a fixed pairwise protocol.** Under the same image and prompt, the annotator was asked to judge only one thing: **which response was better grounded in the image** under the same rubric. If the difference was unclear, the pair was marked as ambiguous rather than forced into a binary decision. We will clarify this protocol in the revision for reproducibility.
> > >
> > > > Clarifying the advantage of IRIS over OPA-DPO
> > >
> > > We would like to clarify that the key difference from OPA-DPO lies in how preference alignment is performed. While OPA-DPO relies on external feedback to guide preference construction, IRIS performs alignment using the model’s own internal signals. This gives IRIS **three** main advantages:**(1)** it does not rely on external annotators or GPT-4V-style feedback during preference alignment; and **(2)** its SFT stage serves only as an initial calibration of the implicit reward landscape, after which on-policy pairs are generated and optimized directly, whereas OPA-DPO needs an additional SFT-based alignment step to make externally revised responses learnable. **(3)** it is lightweight and scalable.
> > >
> > > Empirically, IRIS remains competitive on hallucination benchmarks while better preserving coverage.
> > >
> > > > Why static human-labeled data are not sufficient after SFT
> > >
> > > We respectfully disagree with the premise that SFT eliminates the need for newly generated pairs.
> > >
> > > We would like to **first clarify one technical point**. SFT and DPO **require different supervision formats: SFT uses single targets, while DPO requires preference pairs.** Therefore, the presence of human-labeled SFT data is not a substitute for the pairwise supervision required by DPO.
> > >
> > > More importantly, **the key issue is not simply whether** **SFT** **has shifted the policy, but whether a fixed human-labeled dataset can continue to provide informative pairwise supervision as the policy evolves.** In our setting, static human-labeled data may still be useful at the beginning, but they **do not necessarily remain well matched to the model’s later-stage failure modes.**
> > >
> > > This matters especially for DPO, since the optimization **is most useful when it distinguishes among the model’s own plausible outputs under the current policy.** Once the policy changes after SFT, the remaining errors are increasingly specific to that updated policy, rather than to the original labeled set.
> > >
> > > For multimodal hallucination, this issue becomes even more pronounced. The remaining errors are often cases where language priors still override visual evidence, and **these policy-specific grounding failures are exactly what newly generated on-policy pairs are designed to capture.** This is exactly what we observe in Table 3: **replacing static human pairs with self-generated pairs** leads to faster and more sustained gains across rounds, with substantially better Round-2 results.
> > >
> > > > Why on-policy generated pairs remain useful, controllable, and scalable
> > >
> > > We agree that newly generated pairs should not be assumed to be perfect. However, under DPO, **usefulness does not require every pair to be absolutely correct in a human-gold sense; what matters is whether the constructed preference direction provides a useful relative learning signal.**
> > >
> > > More importantly, **IRIS does not optimize on raw self-generated outputs.** The sampled candidates are first sifted in the model’s own implicit-reward space, where RVG explicitly down-weights responses that are more strongly supported by language priors than by visual evidence. We further reduce noisy supervision through screening, length-aware filtering, and conservative anchoring before optimization. **The generated pairs are therefore filtered supervision, not uncontrolled noise.**
> > >
> > > Finally, the scalability of IRIS **does not rely on manual inspection.** As stated earlier, the manual check was included only as an additional human verification in the rebuttal, rather than as part of the training pipeline itself. Pair construction is fully built into the automatic sampling-and-sifting pipeline, and all filtering steps are internal to training. Consistently, **Table 5 shows that increasing the on-policy preference data from 1k to 5.7k continues to improve performance rather than destabilize training.**

---

### Official Review · Reviewer_RWDu · 2026-03-11

**Soundness:** 3
**Presentation:** 3
**Significance:** 2
**Originality:** 3
**Overall Recommendation:** 4
**Confidence:** 4

**Summary:**

This work studies the hallucination problem of MLLMs from the perspective of DPO optimization. The authors identify critical issues in existing methods—specifically, the off-policy learnability gaps and discretization loss stemming from a heavy reliance on external evaluators. To address these, they propose IRIS, a framework that leverages continuous implicit rewards within the native log-probability space for DPO. This approach effectively mitigates language priors and enhances the model's grounding capabilities. The comprehensive experimental results validate the effectiveness and robustness of the proposed IRIS method.

**Compliance With Llm Reviewing Policy:**

Affirmed.

**Final Justification:**

Most of my concerns have been addressed. However, I still think that the warm SFT stage contributes substantially to the final results, while this stage itself is not the main contribution of the paper. In addition, regarding Weakness 3, a comparison between the external evaluator and the internal confidence score would make the conclusion more convincing.

Overall, I will maintain my original score.

**Key Questions For Authors:**

1. I am interested in a deeper exploration of why external correction leads to distribution shifts. Can the authors provide more detailed quantitative results or a theoretical analysis for this phenomenon?
2. Regarding the penalty term in Eq. (10), if we expand the log-difference, it can be interpreted as the ratio of the visual grounding capability of the reference model versus the current model. Therefore, can this penalty term be viewed as a dynamic constraint based on the relative visual grounding strength?

**Limitations:**

The authors should provide a more comprehensive discussion on the operational boundaries of implicit rewards. Please specify the types of multimodal scenarios or data distributions where implicit signals are most effective, and conversely, identify cases where they might fail to provide sufficient discriminative power compared to explicit labels.

**Strengths And Weaknesses:**

**Strength:**

1. The manuscript is exceptionally well-written. The logical flow is coherent, the motivation is high-level, and the overall structure is rigorous and complete.
2. The derivation of implicit rewards from the native log-probability space is theoretically sound and offers an elegant solution.
3. The authors provide extensive evaluations that demonstrate the superior performance and robustness of IRIS across multiple benchmarks.

**Weaknesses:**
1. In the Introduction, the design of the preference signal appears to encompass two distinct stages: the construction of preference answers and the scoring of reference pairs. For clarity, it would be beneficial to explicitly distinguish between the "preference answer" and the "preference signal" in lines 55-62. This distinction would better contextualize the inherent limitations of external evaluators discussed later.
2. The authors employ an SFT warmup phase prior to IRIS. Since SFT is inherently an off-policy optimization, it typically introduces distribution shifts. This seems to be in tension with the paper’s primary claim of pursuing an on-policy optimization trajectory. Given that the SFT warmup significantly impacts performance (e.g., on Object Hallucination), does this suggest the method is highly sensitive to the initial strength of the base model? Further clarification on this dependency is requested.
3. The assertion in the Introduction that "discrete external rewards assign identical scores" lacks direct empirical support. A quantitative analysis or a toy experiment would strengthen this claim. Furthermore, it is essential to provide a comparative evaluation between the proposed method and a baseline that utilizes "self-generated preference answers scored by external evaluators."
4. The experimental settings and presentation in Table 3 should be more detailed. Specifically, the comparison between the proposed implicit signal and traditional external evaluator-based methods needs to be more explicit. Additionally, reporting results on the LLaVA-13B architecture would provide a more comprehensive view of the method's scalability.
5. The authors should discuss or provide results on how the proposed method performs when applied to different MLLM architectures beyond LLaVA.

---

> ### Author Rebuttal · Authors · 2026-03-31
>
> Dear Reviewer RWDu,
>
> We thank you for the positive assessment and for describing our work as "exceptionally well-written" and "theoretically sound." We address your questions below.
>
> **Response to Weakness 1:**
>
> We will clarify in the Introduction that preference answers and the preference signal are distinct in IRIS: candidate answers are first sampled from the current policy, and implicit rewards with RVG are subsequently used as the signal to score and sift them into preference pairs.
>
> **Response to Weakness 2:**
>
> We agree that SFT warm-up is off-policy, and we will clarify that our on-policy claim refers specifically to the iterative IRIS alignment rounds after initialization. In these rounds, preference pairs are constructed from responses sampled from the current policy. In this sense, SFT serves as a **warm-start calibration** for the initial implicit reward landscape.
> Regarding sensitivity, **Table 3** suggests that IRIS benefits from a **minimally calibrated starting point**, but is not simply driven by the initial model strength. On Object HalBench (**CHAIRs**), SFT alone reaches **17.3** at R2, self-generation alone reaches **18.3**, while combining both yields **8.6**. This indicates that **SFT and on-policy self-generation are complementary**: SFT improves the starting regime, while the substantial gains come from the subsequent on-policy rounds. We will revise the paper to clarify this dependency more explicitly.
>
> **Response to Weakness 3:**
>
> We agree that this point should be supported with a direct experiment. In the revision, we added a direct study of the exact baseline under discussion: **self-generated candidate answers scored by an external evaluator**. For **5,733 prompts**, we used the **28,665 candidate responses generated at the R0 stage by LLaVA-1.5-7B** (i.e., **five candidates per prompt**) and scored them with **Qwen3-VL-30B** using discrete rewards. Under a **1–3 scale**, the **pairwise tie rate** *(same-score rate over within-prompt candidate pairs)* is **94.47%**, and the **all-tie rate** *(all five candidates under a prompt receive the same score)* is **15.16%**. Even under a **1–5 scale**, the pairwise tie rate remains **86.27%**, with frequent ties at both the highest and lowest scores. This provides direct support for our motivation that **discrete external rewards often fail to distinguish self-generated candidates**, making preference construction ambiguous.
>
>
> **Response to Weakness 4:**
>
> We agree that the current presentation of **Table 3** can be clearer, and we will update both the **experimental setting description** and the **table presentation** in the revision.
>
> More specifically, we have added a direct baseline where **self-generated candidates are scored by an external evaluator using discrete 1–3 labels to construct preference pairs**, and then **train the DPO model with the same model and optimization settings as IRIS**. :https://anonymous.4open.science/r/code-D288/table/Comparison.png
>
> Regarding scalability, LLaVA-13B results are already reported. We will make them more prominent in the revision.
>
> **Response to Weakness 5:**
>
> To demonstrate cross-architecture generalization, we evaluated IRIS on **Qwen2.5-VL-3B**.:https://anonymous.4open.science/r/code-D288/table/Effect.png
>
> **Response to Questions 1:**
>
> The distribution shift from external correction mainly comes from **support mismatch** between the correction signal and the model's current policy.
> First, under **KL-constrained DPO**, optimization is local to the reference policy. External corrections produced by stronger models or human annotators may lie outside the model's high-probability region, so moving toward them can be penalized by the KL constraint and becomes difficult to realize through stable updates.
> Second, from the gradient form in **Eq. 5**, the update becomes weak when the preferred response has very low probability under the reference policy. In this case, even if the external correction is directionally desirable, it is not very learnable for the current model.
> IRIS avoids this issue by constructing preference pairs from **on-policy self-generated responses**, which remain within the model's own support and are therefore more learnable under iterative DPO.
>
> **Response to Questions 2:**
>
> We thank the reviewer for this insightful observation. Expanding the penalty term shows that it reflects a **relative grounding difference** between the current policy and the reference policy. In this sense, it can indeed be viewed as a **dynamic constraint** based on relative visual grounding strength.
> Intuitively, the term penalizes cases where a response becomes better supported under the **text-only** context than under the **image-conditioned** context, relative to the reference policy. This helps suppress responses driven more by **language priors** than by **visual evidence** during iterative training. We will clarify this interpretation in the revision.

---

> > ### Author Rebuttal · Reviewer_RWDu · 2026-04-04
> >
> > Thanks for your response. Most of my concerns have been addressed. However, I still think that the warm SFT stage contributes substantially to the final results, while this stage itself is not the main contribution of the paper. In addition, regarding Weakness 3, a comparison between the external evaluator and the internal confidence score would make the conclusion more convincing.
> >
> > Overall, I will maintain my original score.

---

> > > ### Author Response · Authors · 2026-04-08
> > >
> > > Dear Reviewer RWDu,
> > >
> > > Thank you for your careful reading and helpful suggestions. We respond to your remaining points below.
> > >
> > > >However, I still think that the warm SFT stage contributes substantially to the final results, while this stage itself is not the main contribution of the paper.
> > >
> > > Thank you for this helpful observation. We agree that the warm SFT stage contributes substantially to the final performance, but we do not claim it to be the main contribution of the paper. **Its role is to provide a stronger initialization for the subsequent IRIS procedure**.
> > >
> > > | Warm-start setting          | chair ↓ |    hal ↓ |   cog ↓ | CHAIRs (%) ↓ | CHAIRi (%) ↓ |
> > > | --------------------------- | ------: | -------: | ------: | -----------: | -----------: |
> > > | 1k SFT                      |     5.2 |     22.7 |     2.7 |         32.0 |        16.37 |
> > > | **1k SFT + 1 round IRIS**   | **3.9** | **17.4** | **2.0** |    **17.67** |     **9.56** |
> > > | 3k SFT                      |     5.9 |     28.7 |     3.1 |        24.67 |        12.32 |
> > > | **3k SFT + 1 round IRIS**   | **4.2** | **20.0** | **2.2** |     **17.0** |     **8.47** |
> > > | 5.7k SFT                    |     5.3 |     25.5 |     2.5 |         24.0 |        13.04 |
> > > | **5.7k SFT + 1 round IRIS** | **3.8** | **17.5** | **1.6** |     **17.3** |     **8.45** |
> > >
> > > To clarify this point, we additionally varied the warm-start strength and then applied the same one-round IRIS update on top of each model. As shown in the table, IRIS consistently brings further gains over the corresponding SFT-only model across all three settings. We believe this supports a more precise conclusion: **warm SFT is an important initialization step, while the main contribution of IRIS is the additional improvement brought by the subsequent on-policy self-evolution.**
> > >
> > >
> > >
> > > >In addition, regarding Weakness 3, a comparison between the external evaluator and the internal confidence score would make the conclusion more convincing.
> > >
> > > Thank you for this suggestion. We agree that **a more direct comparison between the external evaluator and the internal score would make this point clearer.** In our previous rebuttal, we **already added** the baseline based on self-generated answers scored by an external evaluator. Here we **further compare** the two signals **on the final preference pairs used by IRIS.**
> > >
> > > Specifically, we aligned the external discrete scores with all final preference pairs constructed by IRIS.
> > >
> > > - We find that in 2,339 of these pairs (40.8%), the external evaluator gives exactly the same score to both responses. This means that even at the final pair level, the external evaluator **often cannot provide a clear preference signal.**
> > >
> > > - By contrast, on these externally tied pairs, the internal reward used by IRIS still shows a clear continuous gap **(mean ΔS = 0.0107, std = 0.0056)**. We believe this directly supports our point: **the value of the internal score is that it preserves useful ranking information exactly where discrete external scoring becomes too coarse.**
> > >
> > >
> > > We thank the reviewer again for the careful reading and constructive suggestions. We hope that the additional analyses and clarifications above have adequately addressed the remaining concerns.

---

### Official Review · Reviewer_FSHw · 2026-03-12

**Soundness:** 4
**Presentation:** 3
**Significance:** 2
**Originality:** 3
**Overall Recommendation:** 4
**Confidence:** 4

**Summary:**

This paper proposes IRIS (Implicit Reward-Guided Internal Sifting), an on-policy preference alignment method to reduce multimodal hallucinations in MLLMs. Instead of using external evaluators (e.g., GPT-4V) to score or rewrite responses, which introduces discretization and off-policy learnability issues, IRIS uses the model’s own implicit rewards (log-probability ratios between current and reference policies) to build preference pairs. After an SFT warm-up on 5.7k samples, the method iteratively samples multiple responses per prompt, scores them with Rectified Visual Guidance (RVG), which compares image-conditioned vs. text-only log-ratios to downweight language-prior-driven answers, and trains with a combined DPO-style objective (conditional textual and visual preference plus anchored regularization). With no external feedback during alignment, IRIS achieves strong results on AMBER, Object HalBench, and MMHal-Bench (e.g., large drops in CHAIR and HalRate vs. base LLaVA-1.5) and is competitive with or better than baselines that use larger datasets and external evaluators, while reducing preference-data curation cost by roughly an order of magnitude compared to prompt-based labeler pipelines.

**Compliance With Llm Reviewing Policy:**

Affirmed.

**Key Questions For Authors:**

1. How does SFT warm-up training affect DPO performance, particularly in the context of reducing hallucinations? As noted in the paper, a better SFT model provides a stronger implicit reward and can be used to generate higher-quality preference pairs. I am interested in understanding more specifically how the SFT phase contributes to mitigating hallucination in the subsequent DPO stage.

2. The paper states: "Under the delta learning view (Geng et al., 2025), learning can still progress as long as the induced preference direction is correct more often than not." This connects closely to my first question. How can we determine how much SFT training is sufficient to ensure that the induced preference direction is indeed "correct more often than not"? In other words, what criteria indicate that the SFT model has reached a point where this assumption becomes valid?

3. If the SFT model is already well-trained to the extent that hallucinated samples are rarely generated and thus difficult to source as negative examples, does this create a practical challenge for constructing effective preference pairs for DPO?

4. Why do you choose these base models?

**Limitations:**

1. Lemma E.1 seems an existing results from previous papers. It should not be considered as the original contribution for this paper.

**Strengths And Weaknesses:**

**Soundness:**

**Strength:**
1. The method is thoroughly evaluated against a comprehensive set of state-of-the-art baselines.
2. Experiments are conducted on both 7B and 13B models, and the consistent results effectively demonstrate the superior performance of IRIS.
3. The approach is supported by extensive ablation studies, covering training settings, hyperparameters, and other critical components.

**Weakness:**
1. The base model used is somewhat outdated.

**Presentation:**

**Strength:**
1. The paper is generally well-structured and a pleasure to read. The methodology is presented in a logical, step-by-step manner that is easy to follow.
2. Necessary details are provided in the appendix, allowing readers to gain a more in-depth understanding of the work.

**Weakness:**
1. There are minor misalignments in the notations. For example, the paper states, "In each round r, the reference policy πref is a frozen copy of the preceding policy πθr−1," which conflicts with another description: "and the reference policy π(r−1)ref , taken as the preceding policy πθr−2."

**Significance:**

**Strength:**
1. The paper addresses a critical problem in the field of Multimodal Large Language Models (MLLMs): hallucination.

**Weakness:**
1. While the problem is significant, the core idea has been extensively explored in the text domain, most notably by:
    - Chen et al. 2024a – DICE (ICLR 2025) “Bootstrapping Language Models with DPO Implicit Rewards”
    - **Similarities to IRIS:** DICE also utilizes the DPO implicit reward (the log ratio between the current and reference policy) to construct preference pairs without external feedback. It achieves this by sampling multiple responses, scoring them with the implicit reward, and using these scores to create preference data for iterative DPO. DICE also incorporates length-normalized scoring and on-policy, self-generated data.
    - **Differences:** DICE is designed for text-only LLMs and instruction following. It does not leverage multimodal inputs (e.g., comparing responses with and without an image) and lacks a mechanism like IRIS's RVG for explicit visual grounding. While IRIS successfully extends this line of work to the multimodal domain, the introduction of RVG, in my opinion, represents an incremental contribution rather than a sufficiently novel one.

**Originality:**

**Strength:**
1. The RVG (Rectified Visual Grounding) design is a clever and effective mechanism for down-weighting samples that do not adequately rely on visual information.
2. To the best of my knowledge, this paper is the first to extend the DICE-like methodology from the text-only domain to the multimodal domain.

**Weakness:**
1. The overall concept of the work is highly similar to DICE (ICLR 2025) “Bootstrapping Language Models with DPO Implicit Rewards,” which limits its originality.

---

> ### Author Rebuttal · Authors · 2026-03-31
>
> Dear Reviewer FSHw,
>
> We sincerely thank you for the positive assessment of our **experimental thoroughness**, **presentation clarity**, and the **effectiveness of our RVG design**. We appreciate your constructive feedback and address your specific concerns below.
>
> ---
>
> ****Soundness** Response to Weakness 1 and Question 4:**
>
> We chose **LLaVA-1.5** because it is a **widely used open benchmark model** in prior hallucination-mitigation work, which enables **direct and fair comparison**. Our evaluation is also not limited to a single setting: the paper includes both **7B and 13B** results, and we further add **Qwen2.5-VL-3B** results showing consistent gains.
> (results available at:https://anonymous.4open.science/r/code-D288/table/Effect.png)
>
> ---
>
> ****Presentation** Response to Weakness 2:**
>
> We thank the reviewer for pointing this out. We agree the notation can be clearer, as the term "reference policy" is reused for two distinct roles. In Sec. 4.3, it denotes the scoring reference for RVG sifting ($\pi_{\theta_{r-2}}$, or $\pi_{\theta_{\mathrm{base}}}$ for $r=1$); in Sec. 4.4, it denotes the standard frozen DPO reference within round $r$ ($\pi_{\theta_{r-1}}$). We will make this distinction explicit in the revision.
>
> ---
>
> ****Significance**  Response to Weakness 3 and Weakness 4:**
>
> We thank the reviewer for recognizing RVG’s effectiveness and the value of extending this line of work to the multimodal setting.
>
> Regarding DICE, we agree that our work shares a similar high-level idea of self-alignment, as discussed in our Related Work section.
>
> However, we believe IRIS is more than a direct extension of DICE, because it is designed for multimodal hallucination, where the key challenge is to separate visual evidence from language priors.
>
> In DICE, a single implicit reward is used to rank responses for text-only alignment. In IRIS, the challenge is different: whether a response is actually supported by the image, rather than only by language priors. For this reason, IRIS reformulates implicit rewards into two signals for the same response: an image-conditioned reward and a text-only reward. Under this setting, RVG is a natural way to identify responses that are driven more by language priors than by the image, and to downweight them during pair construction.
>
> This is also supported by Table 4, where RVG consistently gives lower hallucination than using either score alone.We will revise the paper to clarify this distinction.
>
> ---
>
> **Response to Question 1-3:**
>
> We thank the reviewer for these related questions. They all concern the role of SFT warm-up in enabling effective self-generated preference learning in IRIS. Our key point is that SFT is not the main source of final hallucination reduction; rather, it serves as a calibration stage that shapes the model’s implicit reward toward visual grounding, thereby making subsequent preference optimization more reliable.
>
> 1.The SFT phase contributes to hallucination mitigation primarily through value calibration. In untuned MLLMs, implicit rewards are often dominated by text-only priors, so high likelihood may reflect linguistic plausibility rather than true visual grounding. SFT helps align the model’s internal scoring with visual evidence, making probability differences more informative for distinguishing visually supported responses from plausible but unsupported ones. This calibration is important for IRIS, since it enables more reliable construction of preference pairs and provides the directionally correct signal needed for subsequent DPO to suppress language-prior-driven hallucinations.
>
> 2.We do not view sufficient SFT as a fixed stopping threshold. Rather, SFT becomes sufficient once it induces a reliable grounded ranking signal, such that visually grounded responses are ranked above hallucinated or language-prior-driven ones more often than not. At that point, the resulting preference signal becomes directionally useful for subsequent DPO, even if the warm-started model remains imperfect.
>
> 3.If the warm-started model becomes stronger and hallucinated responses become less frequent, constructing highly informative preference pairs may indeed become harder. However, IRIS does not require abundant grossly hallucinated negatives; it only requires sufficient relative contrast within the current policy distribution. By sampling multiple candidates and selecting the most versus least grounded ones using RVG, IRIS can still identify informative pairs even when obvious hallucinations become rarer. We therefore view this as a potential late-stage reduction in pair informativeness, rather than a fundamental limitation of the method.
>
> ---
>
> **Response to Limitation:**
>
> We agree that Lemma E.1 is a standard derivation, included only to keep the theoretical analysis self-contained. Our paper-specific theoretical contributions lie in the subsequent analyses in Secs. E.3 and E.4, which are tailored to the IRIS setting.

---

> > ### Author Rebuttal · Reviewer_FSHw · 2026-04-05
> >
> > Thanks, my concerns are addressed.

---

> > > ### Author Response · Authors · 2026-04-08
> > >
> > > Dear Reviewer FSHw,
> > >
> > > Thank you very much for your careful reading and thoughtful feedback throughout the review process. We are glad that our rebuttal addressed your concerns, and we sincerely appreciate your positive assessment of our work.
> > >
> > > If you feel the rebuttal has fully resolved the key issues, we would be very grateful if you would consider reflecting this in your final score. Thank you again for your time and support.

---

### Decision · Program_Chairs · 2026-04-30

**Decision:**

Accept (regular)

**Comment:**

I find this paper to be a borderline accept / weak accept: it addresses an important problem, proposes a coherent method that combines implicit-reward-based on-policy preference construction with RVG for better visual grounding, and presents solid empirical gains plus meaningful ablations, including evidence that IRIS continues to improve over SFT-only warm starts and that internal continuous scores retain discriminative power where external discrete scorers often tie. At the same time, the main weaknesses raised by reviewers are real: the novelty is moderate rather than high, the core evidence is still concentrated on the LLaVA-1.5 family, and the “on-policy” framing should be stated more carefully given the SFT initialization. After rebuttal, however, two reviewers remained at weak accept, with one explicitly stating their concerns were fully addressed, while the remaining negative reviewer’s concerns center more on scope and validation expectations than on a fatal flaw. On balance, I view the paper as technically sound and likely useful to the community, but with limitations in generalization and novelty that keep it near the threshold; therefore, my recommendation is accept.